# Protein thermal sensing regulates physiological amyloid aggregation

Dane Marijan[1,2], Evgenia A. Momchilova[1,2,8], Daniel Burns [3,8], Sahil Chandhok[1,2,8], Richard Zapf[1,2], Holger Wille [4,5,6], Davit A. Potoyan [3,7] & Timothy E. Audas[1,2] ✉

To survive, cells must respond to changing environmental conditions. One way that eukaryotic cells react to harsh stimuli is by forming physiological, RNA-seeded subnuclear condensates, termed amyloid bodies (A-bodies). The molecular constituents of A-bodies induced by different stressors vary significantly, suggesting this pathway can tailor the cellular response by selectively aggregating a subset of proteins under a given condition. Here, we identify critical structural elements that regulate heat shock-specific amyloid aggregation. Our data demonstrates that manipulating structural pockets in constituent proteins can either induce or restrict their A-body targeting at elevated temperatures. We propose a model where selective aggregation within A-bodies is mediated by the thermal stability of a protein, with temperature-sensitive structural regions acting as an intrinsic form of post-translational regulation. This system would provide cells with a rapid and stress-specific response mechanism, to tightly control physiological amyloid aggregation or other cellular stress response pathways.

One of the fundamental features of life is the ability to detect and respond to changing cellular conditions. This feature is critically important, as deviations from a normal growth environment occur frequently, and modulation of metabolic processes is necessary to maintain homeostasis. Collectively, these adaptations are known as stress response pathways, with cell survival depending on sensing the correct stress and eliciting an appropriate response. Stimuli such as heat shock[1], DNA damage[2], hypoxia[3], oxidative stress[4], and unfolded protein accumulation in the ER[5] each require unique metabolic changes to facilitate cell recovery. However, significant crosstalk between signaling cascades often occurs, as disparate stimuli can activate common elements of certain stress response pathways. For example, heat shock proteins are upregulated by elevated temperatures, low oxygen availability[6], heavy metal contamination[7], and oxidative stress[8].

Thus, gaining insight into the molecular mechanisms that a cell uses to activate specific and common stress-response factors is crucial to understanding these important signaling networks.

Over the last decade, the emergence of biomolecular condensates has altered the landscape of cellular stress response research. Across the eukaryotic domain, the formation of membrane-less organelles have been tightly linked to changing environmental conditions[9,10]. In yeast, energy depletion and low pH cause reversible Sup35 condensate formation[11], while Sec bodies of *Drosophila melanogaster* form during amino acid starvation[12]. Mammalian nuclei are a hub for stress-induced condensate biogenesis[13], with foci such as paraspeckles, nuclear stress bodies, and nucleolar aggresomes forming in response to an array of cellular insults[14–18]. Though much remains to be learned about the functionality of these condensates, common biological effects of their

[1]Department of Molecular Biology and Biochemistry, Simon Fraser University, 8888 University Drive, Burnaby, BC V5A 1S6, Canada. [2]Centre for Cell Biology, Development, and Disease, Simon Fraser University, 8888 University Drive, Burnaby, BC V5A 1S6, Canada. [3]Roy J. Carver Department of Biochemistry, Biophysics and Molecular Biology, Iowa State University, Ames, IA 50011, USA. [4]Department of Biochemistry, University of Alberta, Edmonton, Alberta T6G 2H7, Canada. [5]Centre for Prions and Protein Folding Diseases, University of Alberta, Edmonton, Alberta T6G 2M8, Canada. [6]Neuroscience and Mental Health Institute, University of Alberta, Edmonton, Alberta T6G 2E1, Canada. [7]Department of Chemistry, Iowa State University, Ames, IA 50011, USA. [8]These authors contributed equally: Evgenia A. Momchilova, Daniel Burns, Sahil Chandhok. ✉e-mail: taudas@sfu.ca

formation include the modulation of biochemical reactions and the buffering of cellular protein concentration[19]. Biophysically, many of these condensates possess liquid-like properties, as proteins sequestered within these structures retain much of their dynamic profiles[20]. The transition of these condensates from a liquid to solid state has been linked to numerous disorders, as striking evidence suggests that aberrant aggregation of disease-associated proteins within these structures may underlie the etiology of prominent human pathologies[10,20]. Unlike their liquid-like counterparts, stress-inducible amyloid bodies (A-bodies) were found to naturally possess solid-like properties, as a heterogeneous family of proteins adopt an amyloid-like structure within this subnuclear domain[21,22]. The amyloid state is characterized by unbranching fibrils of polypeptides stacked perpendicularly to form a β-strand-rich, insoluble, and immobile amyloid spine[23]. Although this protein form has been historically viewed as toxic[24], there is growing evidence that eukaryotic cells utilize functional amyloids in numerous physiological roles, including the storage of peptide hormones[25], melanin synthesis[26], sequestration of organelles and RNA in oocytes[27], programmed necrosis signaling[28], and memory persistence[29,30]. Biogenesis of A-bodies occurs in response to elevated temperature or low environmental pH, while a subsequent return to normal growth conditions results in their disassembly[21,31]. These stimuli mediate the expression of low complexity non-coding RNA derived from the ribosomal intergenic spacer region, which nucleates the formation of dense, fibrillar, protease-resistant, and Congo red-positive condensates[21,32,33]. As these reversible amyloids have been found in species from across the eukaryotic domain[31], the evolutionary conservation highlights the significance of this biomolecular condensate and suggests that these foci represent an ancient stress response pathway.

We recently showed that the A-body proteome was dependent on the stressor that induced this physiological amyloid aggregation event[22]. While some proteins acted as universal targets, other constituents are sequestered only in response to specific stimuli. Based on the low complexity nature and non-specific charge-based interactions ascribed to the seeding noncoding RNA, we hypothesized that a protein-level mechanism must exist to discriminate the constituents upon exposure to A-body-inducing stressors. Here, we use two related protein pairs as model systems to identify critical structural elements that mediate heat shock-specific amyloid aggregation. DDX39A and DDX39B are members of the DEAD box family of RNA helicases, which have been implicated in genomic integrity[34], RNA splicing[35], and mRNA export[36], while the heterogeneous nuclear ribonucleoprotein A0 (hnRNPA0) and A1 (hnRNPA1) have been linked to multiple aspects of RNA biogenesis[37]. Thus, stress-specific recruitment of these proteins could significantly alter genome stability and RNA metabolism, providing a site of cellular regulation that tailors this stress-response pathway to different environmental perturbations. Our work demonstrates that individual proteins can possess thermo-sensitive structures that act as direct temperature sensors to mediate A-body recruitment and aggregation. Interestingly, the intrinsically disordered domains of these proteins did not contribute to this stress-specific aggregation mechanism, as the highly-ordered structures possessed the thermo-sensing functionality. Collectively, these data represent a post-translational regulatory mechanism where physiological amyloid aggregation can be rapidly and specifically controlled by the thermo-sensitivity of the tertiary structure of individual cellular proteins.

## Results
### Differential A-body targeting of related RNA helicases during heat stress
The mechanism regulating stress-specific amyloid aggregation within the A-body has not been established, though screening the acidotic[21] and heat shock A-body proteomes[22] revealed a pair of closely related

RNA helicases that possess different targeting specificities. DDX39A and DDX39B share 90% sequence identity (Supplementary Fig. 1a) and a similar dumbbell-shape when comparing the predicted (DDX39A) and crystal (DDX39B) structures[38] (Fig. 1a). However, proteomics data suggests that DDX39A aggregates within both acidotic- and heat shock-induced A-bodies, while DDX39B is only present in the acidosis-induced structures[21,22]. To validate these intriguing mass spectrometry results, we expressed GFP-tagged DDX39A and DDX39B in MCF-7 cells. As predicted, under acidotic conditions both proteins were targeted to Thioflavin S-positive structures that contained the A-body marker molecule CDC73 (Fig. 1b). For these RNA helicases, the putative adoption of an amyloid-like conformation within A-bodies was demonstrated by the hallmark shift towards the insoluble (Supplementary Fig. 1b) and immobile (Supplementary Fig. 1c, d) biophysical state and the inherent capacity of these proteins to generate fibrils, when expressed in a bacterial inclusion body assay (Fig. 1c: a model setting of amyloid aggregation[21,39,40]). Heat shock exposure also induced the expected results, as DDX39A formed aggregates in Thioflavin S- and CDC73- positive foci (Fig. 1b and Supplementary Fig. 1b, c, e), while DDX39B remained nuclear, soluble, and mobile (Fig. 1b and Supplementary Fig. 1b, d, e) at elevated temperatures. The stress-specific targeting/aggregation properties of these proteins were also observed in A549 and HEK293 cell lines (Supplementary Fig. 2a, b) and extended to HIS-tagged variants of both proteins (Supplementary Fig. 2c, d), demonstrating that the cell line and GFP-tag did not mediate this divergence in stress-specific A-body aggregation.

To uncover the mechanism regulating heat shock-specific aggregation, we took advantage of the high structural and sequence similarity of DDX39A and DDX39B (Fig. 1a and Supplementary Fig. 1a). Here, a series of substitution constructs were created where regions of DDX39A were replaced by the corresponding sequences in DDX39B (Fig. 1d-Left Panel). Using an established method for quantifying A-body targeting efficiency (Supplementary Fig. 2e)[31,41], a central segment emerged as a critical differentiator of heat shock-induced protein sequestration (Fig. 1d, e), insolubilization (Supplementary Fig. 2f, g), and immobilization (Fig. 1f). DDX39A constructs containing amino acids 100-250 of DDX39B remained nuclear, soluble and mobile under heat shock conditions, while the presence of amino acids 100-250 from DDX39A targeted any DDX39B construct into the A-bodies, immobilizing and insolubilizing the chimeric protein (Fig. 1d−f and Supplementary Fig. 2f, g). We next wanted to use purified proteins to assess whether DDX39A and DDX39B responded directly to changing temperatures. Unfortunately, inclusion body formation (Fig. 1c) prevented the isolation of bacterially-expressed protein, and multiple attempts to generate soluble mammalian-expressed protein were unsuccessful. Thus, we shifted our efforts to examine the effects of temperature on DDX39 within harvested lysates, which were pre-cleared of aggregates and cellular debris by centrifugation. Soluble supernatants were incubated at 4 °C or 43 °C (1 h), then separated by non-denaturing- and denaturing-PAGE. Despite the uniform presence of protein in the lysates (Fig. 1g, SDS-PAGE: Total), monomeric DDX39B could be seen at both 4 °C and 43 °C, while DDX39A monomers were absent from the 43 °C sample (Fig. 1g, Native-PAGE). To detect putative temperature-induced aggregates, we centrifuged aliquots of the incubated lysates, and ran the decanted pellets on SDS-PAGE. Here, we can see that incubation at 43 °C clearly shifted DDX39A, but not DDX39B, into an aggregated state (Fig. 1g, SDS-PAGE: Pellet), demonstrating that the divergent responses of these proteins to elevated temperatures persist within an in vitro setting. Swapping the central domains (100-250) of these proteins abrogated the thermo-sensitive aggregation of DDX39A, and imparted temperature-mediated insolubilization on DDX39B (Fig. 1g). Together, these results suggested that the central region allows DDX39 proteins to directly sense changing temperatures and mediates stress-specificity amyloid aggregation.

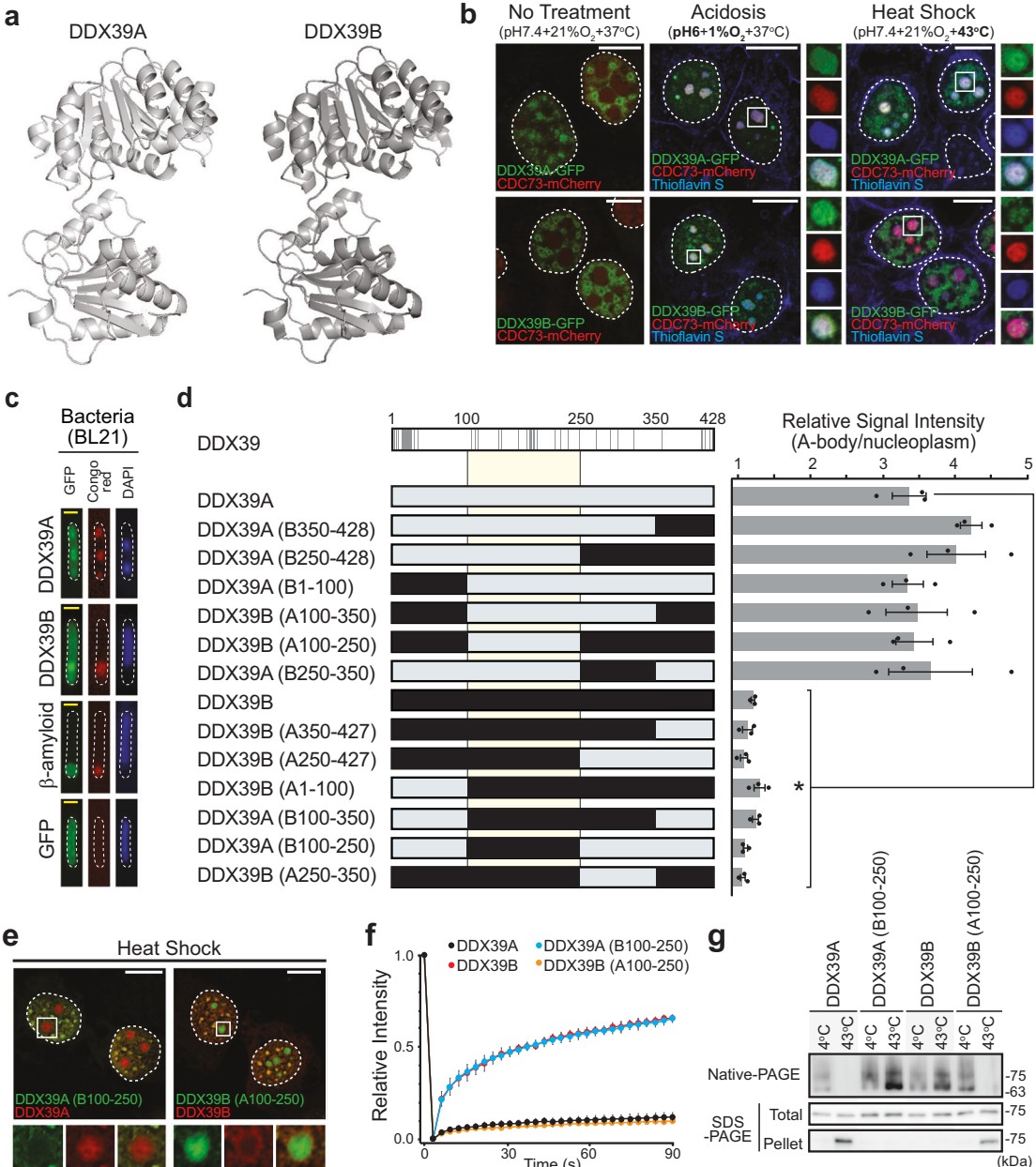

**Fig. 1 | Closely related proteins, DDX39A and DDX39B, are differentially targeted to A-bodies. a** Cartoon 3D structures of DDX39A (predicted by Fold & Function Assignment System) and DDX39B (PDB: 1XTI) (**b**) MCF-7 cells expressing DDX39A-GFP, DDX39B-GFP (green), and the A-body marker protein CDC73-mCherry (red) were left untreated, or exposed to extracellular acidosis (1% $O_2$ + pH6.0) or heat shock (43 °C) for 4 h. Heat shock and acidosis treated cells were stained with the amyloidophilic dye Thioflavin S (blue). **c** DDX39A-GFP, DDX39B-GFP, β-Amyloid-GFP, and GFP were expressed in BL21 cells prior to staining with Congo red and DAPI. Yellow scale bars represent 0.5 μm. **d** Schematic indicating unique residues (gray lines) within the DDX39 proteins (top). The indicated DDX39 substitution constructs (left panel) were expressed in heat shock-treated MCF-7 cells. A-body targeting efficiency was calculated as the average A-body pixel intensity relative to the average nuclear pixel intensity (right panel). For each sample 10 cells were analyzed per replicate, and values represent means ± s.e.m.

($n = 3$ independent experiments, a two-tailed Student's $t$-test was used: *$p \leq 0.05$). **e** DDX39A-mCherry and DDX39A(B100-250)-GFP or DDX39B-mCherry and DDX39B(A100-250)-GFP were co-expressed and visualized in heat shock-treated MCF-7 cells. **f** Quantification of fluorescence recovery after photobleaching data for heat shock-treated (4 h) MCF-7 cells expressing the indicated GFP-tagged DDX39 constructs. For each sample 10 cells were analyzed per replicate, and values represent means ± s.e.m ($n = 3$ independent experiments). **g** DDX39A-GFP, DDX39A(B100-250)-GFP, DDX39B-GFP, and DDX39B(A100-250)-GFP were expressed in MCF-7 cells grown under standard conditions. Soluble lysates were extracted and incubated at 4 °C or 43 °C for 1 h, prior to separation by Native- and SDS-PAGE (Total). Aliquots of the temperature-treated lysates were centrifuged, and aggregates were run on SDS-PAGE (Pellet). Dashed circles represent nuclei, selected regions (white boxes) are expanded (merge: bottom or far-right), white scale bars represent 10 μm. Source data for all graphs and blots are provided with this paper.

## Protein structure regulates a universal A-body targeting motif

Two mechanistic models arise from the data presented above. Either a heat shock-specific A-body targeting motif is present in the central region of DDX39A (absent in DDX39B), or DDX39B possesses a thermo-stable tertiary structure that masks a generic aggregation motif. To test these alternate hypotheses, we created a series of truncation mutations to map the minimal A-body targeting motif. We focused on the N-terminal lobe of the DDX39 proteins, as this domain contains amino acids 100–250 and possesses the same stress-specific A-body targeting phenotypes as the full-length proteins

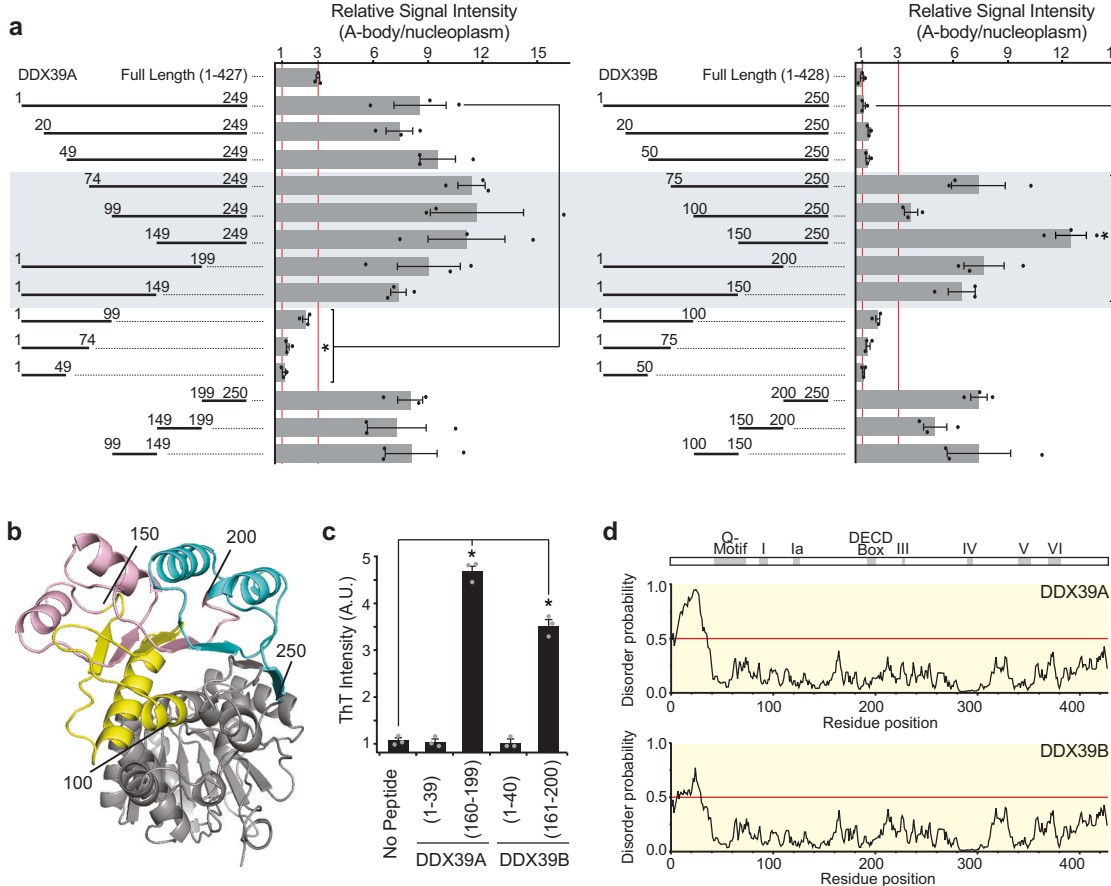

**Fig. 2 | DDX39A and DDX39B contain generic A-body targeting motifs.**
**a** DDX39A (left panel) and DDX39B (right panel) N-terminal truncation mutations were expressed in heat shock-treated MCF-7 cells. A-body targeting efficiency was calculated for each protein fragment. 10 cells were analyzed per replicate, and values represent means ± s.e.m ($n$ = 3 independent experiments, a two-tailed unpaired Student's $t$ test was used: *$p \leq 0.05$). **b** Cartoon 3D structure of DDX39B (PDB: 1XTI) with highlighted minimal A-body targeting motifs: amino acids 100–150 (yellow), amino acids 150–200 (pink), and amino acids 200–250 (cyan). **c** A

Thioflavin T assay was performed with DDX39A (1–39), DDX39A (160–199), DDX39B (1–40), and DDX39B (161–200) peptides. Endpoint (16 h) Thioflavin T fluorescence is presented in arbitrary units (A.U.) as means ± s.e.m. ($n$ = 3 independent experiments, a two-tailed Student's $t$ test was used: *$p \leq 0.05$). **d** IUPred3 disorder prediction maps of DDX39A and DDX39B. A schematic of the generic DDX39 protein and the putative domains is included (above). Source data for all graphs are provided with this paper.

(Supplementary Fig. 3a). Using quantitative microscopy to gauge the efficiency of A-body recruitment (Supplementary Fig. 2e), our results demonstrate that amino acids 75–250 of both proteins were capable of sequestering GFP under heat shock conditions (Fig. 2a). This suggests that DDX39A and DDX39B contain a generic A-body localization domain, which has the capacity to mediate sequestration and aggregation of both proteins at elevated temperatures. Finer mapping showed that there are potentially three subnuclear targeting motifs present within the central region of both proteins (amino acid positions 100–150, 150–200, and 200–250) (Fig. 2a, b and Supplementary Fig. 3b, c). As AmylPred2[42] predicts that each of these regions contain aggregation prone clusters (Supplementary Fig. 3b, c – Bottom Panel), we synthesized DDX39 peptides and ran a Thioflavin T (ThT) fibrillation assay to determine whether DDX39 fragments could adopt an amyloid conformation. In this assay, peptides from the central region of DDX39A (160–199) and DDX39B (161–200) generated ThT-positive amyloid fibrils at a significantly higher rate than AmylPred2-negative DDX39A (1–39) and DDX39B (1-40) regions (Fig. 2c and Supplementary Fig. 3d). Thus, these data point towards DDX39 proteins possessing a generic targeting motif that facilitates the amyloid-like aggregation of these proteins within A-bodies.

This panel of N-terminal truncation mutations also highlights the essential role of amino acids 50–74 in DDX39B stress-specific aggregation. Both DDX39B (50–250) and DDX39B (75–250) contain the

putative A-body targeting motif(s) described above, yet the addition of these 25 amino acid residues prevented heat shock-mediated A-body recruitment (Fig. 2a, Supplementary Fig. 3a). Based on our data, it seemed unlikely that this sequence contains a discrete motif that actively inhibits A-body targeting, as it; (1) falls outside of the stress-specificity region identified in Fig. 1, (2) contains no amino acid substitutions unique to DDX39B (Supplementary Fig. 1a), and (3) fails to disrupt A-body targeting in DDX39B (1-150)/DDX39B (1-200) (Fig. 2a) or DDX39B (100-250) upon direct fusion (DDX39B [50-74 + 100-250], Supplementary Fig. 3e). A more probable explanation for the effects of amino acids 50-74 on A-body stress-specificity, is that this region is important for the proper folding and tertiary structure of this protein. IUPred3 analysis and AlphaFold predictions (AF-Q13838-F1) support this notion, as the first ~50 residues of DDX39B encode an intrinsically disordered domain, while the next 25 amino acids are part of the N-terminal ordered structure (Fig. 2d). Thus, we predict that thermal stability of the tertiary structure could be a critical determinant in heat shock-specific A-body recruitment.

To further characterize the role of the central region of DDX39B in A-body targeting and protein aggregation we generated a panel of single amino acid substitutions that replaced each of the unique residues in DDX39A with their DDX39B equivalents (Fig. 3a). Of the 17 sites that encoded different amino acids within the 100-250 region, multiple substitutions significantly repressed A-body recruitment (Fig. 3b). The

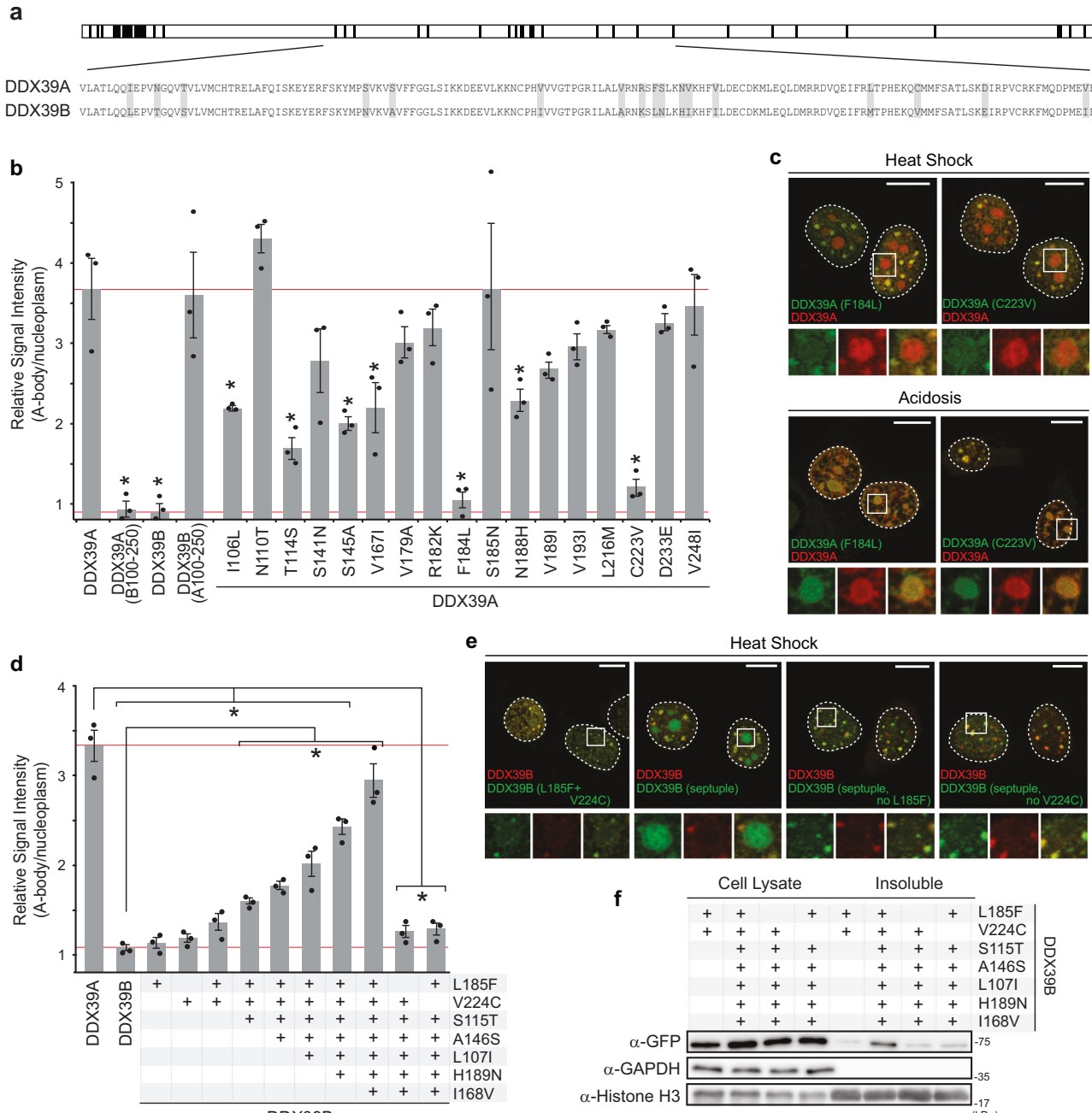

**Fig. 3 | Central amino acid residues can modulate DDX39A and DDX39B heat shock A-body targeting. a** Schematic indicating the positions of unique residues within the DDX39 proteins (black bars, top panel). The sequences of the central region (amino acids 100-250) are provided, with amino acid differences highlighted in gray (lower panel). **b** MCF-7 cells were transfected with mutant DDX39A-GFP constructs containing individual substitutions corresponding to the 17 residue differences in the amino acid 100-250 region. Cells were heat shock-treated and A-body targeting efficiency was calculated. 10 cells were analyzed per replicate, and values represent means ± s.e.m ($n$ = 3 independent experiments, a two-tailed Student's $t$ test was used: *$p \leq 0.05$). **c** DDX39A-mCherry (red) and DDX39A(F184L)-GFP or DDX39A(C223V)-GFP (green) constructs were expressed in MCF-7 cells exposed to heat shock (top) or acidotic (bottom) conditions. Representative images are presented. **d** Heat shock-treated MCF-7 cells expressing the indicated DDX39B

mutations were quantified for A-body targeting. 10 cells were analyzed per replicate, and values represent means ± s.e.m. ($n$ = 3, independent experiments, a two-tailed Student's $t$ test was used: *$p \leq 0.05$). **e** Representative images were taken of the indicated DDX39B-GFP point mutations. DDX39B-mCherry was included as a control. **f** MCF-7 cells transfected with the indicated DDX39B mutants were subjected to heat shock treatment prior to lysate fractionation. Western blotting was used to determine the presence of the indicated DDX39B-GFP constructs in the insoluble fraction, with GAPDH and Histone H3 used as soluble and insoluble controls, respectively. In representative images, dashed circles represent nuclei, selected regions (white boxes) were expanded below (merge: far-right). white scale bars represent 10 μm. Source data for all graphs and blots are provided with this paper.

F184L and C223V substitution were the most prominent, as they could independently reduce A-body localization (Fig. 3b, c) and increase DDX39A mobility (Supplementary Fig. 4a) to DDX39B levels. Under acidic conditions neither F184L nor C223V had any effect on full-

length DDX39A sequestration (Fig. 3c and Supplementary Fig. 4b), and their incorporation into the minimal DDX39A (99-249) constructs also failed to repress A-body recruitment under both heat shock and acidotic conditions (Supplementary Fig. 4c). This suggests that

DDX39B amino acid substitutions are not disrupting the generic acidosis/heat shock A-body targeting motif(s). Next, we sought to perform the reciprocal experiments and impart the heat shock A-body targeting properties of DDX39A on DDX39B. We noted that both F184L and C223V substitutions independently inhibit DDX39A recruitment (Fig. 3b, c), suggesting that these residues exert a dominant-negative effect on this form of heat-induced protein aggregation. Therefore, we expected the presence of either residue in DDX39B would prevent its aggregation. As predicted, the single reciprocal mutations L185F and V224C (DDX39B contains one extra residue at amino acid 18 [Supplementary Fig. 1a], hence the different residue position) failed to induce A-body targeting in DDX39B, as each construct possessed an inhibitory residue at the other established site (Fig. 3d). However, the double substitution DDX39B (L185F + V224C) also did not aggregate within A-bodies at elevated temperatures (Fig. 3d–f and Supplementary Fig. 4a, d), suggesting that other critical inhibitory regions are present within DDX39B. To explore this further, we sequentially added reciprocal amino acid substitutions that were shown to partially impair DDX39A subnuclear targeting (Fig. 3b). Each substitution additively increased heat shock-induced recruitment of DDX39B, culminating in the DDX39B (septuplet: L185F + V224C + S115T + A146S + L107I + H189N + I168V) construct being sequestered, insolubilized, and immobilized within A-bodies as efficiently as DDX39A (Fig. 3d–f and Supplementary Fig. 4a, d). Reverting the L185F or V224C residues back to the wild-type amino acids sequences in the DDX39B (septuplet) construct completely abolished heat shock-induced targeting, insolubilization, and immobilization (Fig. 3d–f and Supplementary Fig. 4a, d). Together, these data generate a putative model where generic A-body targeting motif(s) are masked under heat shock conditions by multiple thermo-stable structural features within the protein.

### Distinct structural pockets regulate heat shock-mediated A-body recruitment

Prompted by these findings, we examined the location of each residue within the predicted structure of DDX39A. This revealed that all seven amino acid substitutions are spatially clustered into three pockets in the N-terminal domain (Fig. 4a). Two of these pockets are hydrophobic in nature, surrounding residues C223 and F184/S145 (Fig. 4b), while the third pocket is hydrophilic and contains the residues T114, I106, N188, and V167 (Fig. 4c). Based on the additive effect of the latter four amino acids on DDX39B sequestration (Fig. 3d–f and Supplementary Fig. 4a, d), we predicted that the combined inhibition of substituting all four residues in DDX39A might parallel the results seen by altering the F184L and C223V pockets (Fig. 3b). Individually, each of these residues impair DDX39A sequestration (Fig. 3b), however, the DDX39A (quadruple; T114S + I106L + N188H + V167I) construct repressed heat shock-induced A-body targeting to wild-type DDX39B levels (Fig. 4d), validating that these four residues cooperatively coordinate a third site of structural regulation in the DDX39 protein. To probe the properties of the hydrophobic pockets further, we replaced the phenylalanine and cysteine residues at positions 184 and 223 of DDX39A, respectively, with the other 19 amino acids and assessed A-body targeting. At position 184 only medium-sized hydrophobic residues (leucine, isoleucine and methionine) were capable of inhibiting A-body targeting (Fig. 4e), while valine was the only amino acid that significantly repressed heat shock-mediated sequestration at position 223 (Fig. 4f). Interestingly, the substitution of large, polar, and/or charged residues at these sites almost uniformly increased A-body targeting beyond the wild-type protein levels (Fig. 4e, f). These findings mirrored the effects observed with large N-terminal truncation mutations, where putatively unstructured fragments displayed greater A-body targeting efficiency than the properly folded full length (1-427 and 1-428) DDX39A and DDX39B (Fig. 2a). We suspected the biophysical properties of the third pocket to be distinct,

as it contains hydrophilic residues. To examine this, we substituted the third pocket residue that shows the greatest inhibitory effect on DDX39A A-body targeting (Fig. 3b: T114), with all 19 other amino acids. The results showed that there was greater variability in the residues that could repress A-body targeting at this location (Fig. 4g), and that charged/polar amino acid substitutions did not dramatically enhance A-body recruitment, unlike the effects seen in the F184L and C223V hydrophobic pockets (Fig. 4e, f). Therefore, these data indicate that structural features regulate A-body targeting, with the three pockets in DDX39B imparting thermo-stability on the protein. Interestingly, the residues comprising the three pockets neatly aligned with each of the three minimal targeting motifs (Supplementary Fig. 5) previously identified (Fig. 2b).

To rigorously challenge this model, our aim was to restore heat shock-induced A-body aggregation to the DDX39A (F184L), DDX39A (C223V), and DDX39A (quadruple) mutant constructs by altering the pockets that surround these critical residues. The selected DDX39A constructs only encode one putative thermo-stable pocket each (i.e., with a DDX39B-like sequence/structure), allowing for precise control of their thermo-stability via strategic introduction of alanine mutations that reduce critical residues to a single methyl group within the three predicted pockets (Fig. 5a: pink, cyan, and yellow). As controls we also designed one alanine mutation for each pocket, V159A, K240A, and N110A, which are adjacent to the rescue substitutions, but whose R groups are oriented away from the respective pocket cores (Fig. 5a: green). Once heat shocked, A-body targeting of DDX39A (F184L), DDX39A (C223V), and DDX39A (quadruple) variants were significantly increased by each of the pocket rescue mutations, while mutants carrying the positional controls remained diffusely distributed (Fig. 5b). Additionally, while rescue variants displayed decreased mobility and solubility when detained in heat shock A-bodies, their analogous positional control mutants remained mobile and soluble (Supplementary Fig. 6a–c). Collectively, these data validate the role of thermo-sensitive structural pockets in heat shock-specific sequestration, allowing the DDX39 tertiary structures to act as intrinsic regulators of their own A-body targeting and aggregation.

### The molecular dynamics of DDX39A are sensitive to elevated temperature

To further investigate the response of DDX39A and DDX39B to heat, we subjected these proteins to a ~1.6 microsecond molecular dynamics simulation at 37 °C and 43 °C. The results validate a model where exposure of A-body targeting motifs likely results from local conformational changes driven by thermally-enhanced dynamics. During the simulation the average root mean square deviation (RMSD) of the ordered regions of the N-terminal lobes of DDX39A and DDX39B (amino acids 46-250) were similar at 37 °C (Fig. 6a, b-top panel). However, at 43 °C the same region of DDX39A exhibited a markedly larger RMSD difference, relative to the DDX39B protein, (Fig. 6a, b-bottom panel), highlighting the thermo-stability of the A-body excluded DDX39B and thermo-sensitivity of the A-body targeted DDX39A at elevated temperatures.

To investigate where these enhanced dynamics are localized, we calculated the root mean square fluctuation (RMSF) of the Cα atoms in each simulation and compared DDX39A and DDX39B at 43 °C. The results clearly demonstrate a large perturbation in dynamics occurring on the N-terminal lobe of DDX39A, centered at serine 183 (Fig. 6c), which constitutes a loop and helix covering the F184 hydrophobic pocket (Fig. 6d). How these dynamics translate to exposure of the A-body targeting motif(s), and how they are altered in the context of the other structural domains is a subject of future investigation. Nevertheless, these results show that the N-terminal domain of DDX39A experiences enhanced dynamics at increased temperature, while DDX39B does not, supporting the overarching hypothesis.

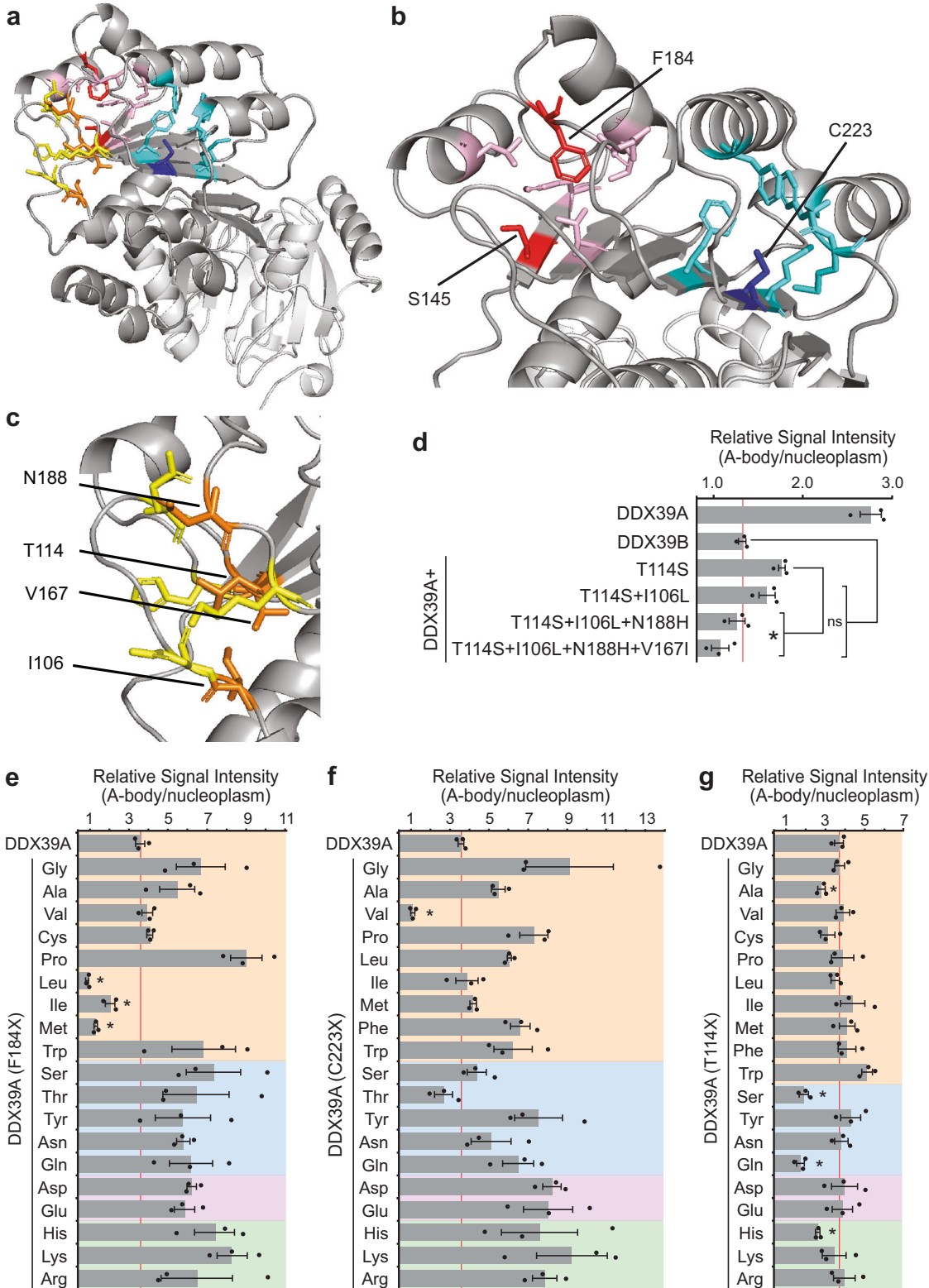

**Fig. 4 | Identification of structural pockets that regulate DDX39 targeting to A-bodies. a** Cartoon of predicted DDX39A structure with residues forming three regulatory pockets highlighted in yellow, red and blue. **b** Enlarged view of the F184 and S145 (red, surrounding residues pink) and C223 (blue, surrounding residues cyan) hydrophobic pockets. **c** enlarged view of the T114, I106, N188 and V167 (orange, surrounding residues yellow) hydrophilic pocket. **d** MCF-7 cells expressing wild-type DDX39A and DDX39B or the indicated DDX39A mutants were subjected to heat shock, visualized, and quantified. Heat shock treated MCF-7 cells expressing constructs containing all possible amino acids at the (**e**) F184X, (**f**) C223X, and (**g**) T114X positions of DDX39A were quantified for A-body targeting efficiency (where X represents amino acids presented along the y-axis of the graph). Background colors indicate amino acid hydrophobicity (orange), polarity (blue), negative charge (purple), or positive charge (green). For each quantification (**d**–**g**), 10 cells were analyzed per replicate, and values represent means ± s.e.m. (*n* = 3 independent experiments, a two-tailed Student's *t* test was used: *$p \leq 0.05$). Not significant (ns). Source data for all graphs are provided with this paper.

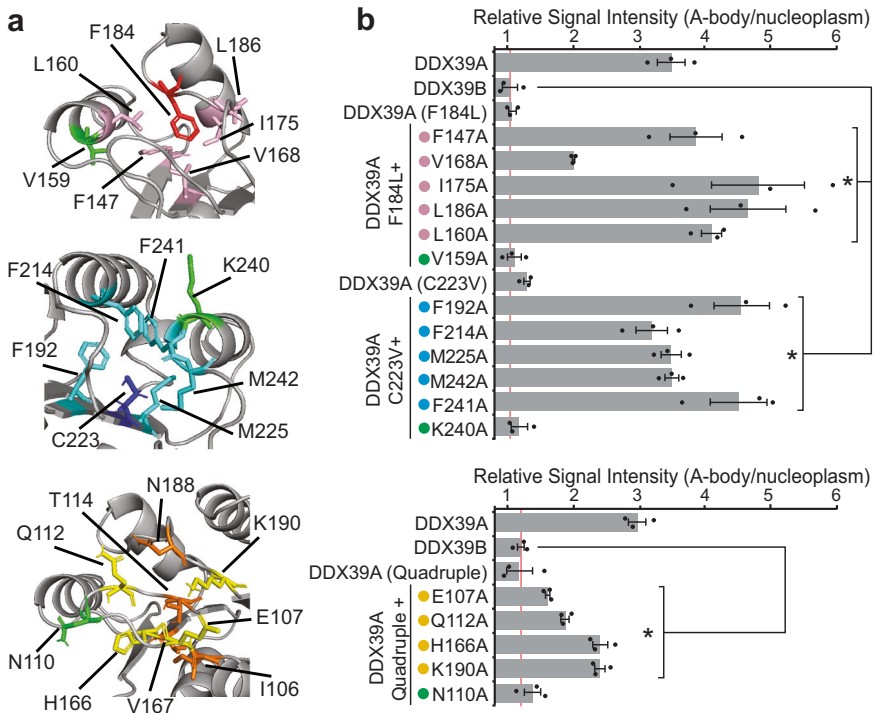

**Fig. 5 | Heat shock A-body targeting is regulated by distinct structural pockets.** **a** Cartoon of the pockets surrounding amino acids F184 (red, top), C223 (blue, middle) and T114/I106/N188/V167 (orange, bottom) in the predicted DDX39A structure. Surrounding residues are shown in pink (top), cyan (middle) and yellow (bottom), while the control residues outside of pockets are indicated in green.

**b** MCF-7 cells expressing the indicated DDX39A mutants were subjected to heat shock conditions and quantified for A-body targeting. 10 cells were analyzed per replicate, and values represent means ± s.e.m ($n = 3$ independent experiments, a two-tailed Student's $t$ test was used: *$p \leq 0.05$). Source data for all graphs are provided with this paper.

## Structural regulation of heat shock A-body recruitment is not exclusive to DDX39 proteins

In order to validate this regulatory mechanism beyond the DDX39 helicases, we searched the MS datasets[21,22] to find related proteins outside the DDX family that may have a heat shock A-body targeting and aggregation pattern akin to DDX39A and DDX39B. This revealed two structurally similar proteins from a heterogeneous nuclear ribonucleoprotein subfamily, hnRNPA0 and hnRNPA1. Both proteins possess two N-terminal globular domains and a large C-terminal IDR (Fig. 7a), with the MS data indicating that only hnRNPA0 is a putative A-body resident during heat shock exposure. By using specific antibodies to visualize the two endogenous proteins, we found that hnRNPA0 and hnRNPA1 acted as expected. In heat shock treated cells, hnRNPA0 was targeted to A-bodies (Fig. 7b) and insolubilized (Supplementary Fig. 7a), while hnRNPA1 remained diffuse (Fig. 7b) and soluble (Supplementary Fig. 7a). Our in vitro assay also showed that hnRNPA0 underwent temperature-sensitive aggregation in extracted cell lysates, while hnRNPA1 remained primarily monomeric and soluble at high temperatures (Fig. 7c). Thus, based on the A-body targeting discrepancy and overall structural similarity we attempted to manipulate the aggregation properties of hnRNPA0 and hnRNPA1 in a fashion analogous to the DDX39A/B experiments above. We confirmed that the GFP-tagged constructs maintain the same heat shock targeting/aggregation properties as their endogenous counterparts (Fig. 7b and Supplementary Fig. 7b) and created chimeric hnRNPA constructs where we substituted the analogous globular domains and/or IDRs (Fig. 7d, e). During heat shock, only constructs carrying the first globular domain of hnRNPA0 (amino acids 1-92) displayed A-body targeting, while those carrying amino acids 1–99 of hnRNPA1 remained diffuse (Fig. 7e). This strongly suggests that regulation of A-body targeting and protein aggregation was mediated by this globular domain of hnRNPA0/A1, and like DDX39, was independent of the intrinsically disordered regions, which were shown not to affect localization

(Fig. 7e and Supplementary Fig. 7c) or mobility during heat shock exposure (Supplementary Fig. 7d). Observation of this regulatory domain suggests that it may represent a simplified version of the DDX39 structure (Figs. 4b and 7f), as this globular region of hnRNPA0/A1 contains a single hydrophobic pocket (i.e., a β-sheet and two helices surrounding a hydrophobic core). To test whether disruption of this hydrophobic core can induce heat shock-mediated A-body recruitment of hnRNPA1, we substituted a central phenylalanine (Fig. 7f) for a destabilizing, charged aspartate residue, strategically choosing a mutation site analogous to the L185 position that was crucial for inducing DDX39B targeting (Fig. 3). As an indirect comparison we also generated L185D or V224D substitutions in DDX39B, before observing protein localization in heat shock treated cells (Supplementary Fig. 7e). Introducing the single destabilizing mutations into either of the DDX39B pockets resulted in A-body recruitment comparable to that of the DDX39B (Septuplet) (Supplementary Figs. 7e, 3e). Remarkably, the analogous F34D mutation in hnRNPA1 also resulted in a gain of heat shock A-body targeting/immobility (Fig. 7e and Supplementary Fig. 7d, e). Finally, to make sure this was not too drastic a destabilization effect, due to a charge residue being introduced into the hydrophobic core, we generated an additional hnRNPA1 mutant with a F34A substitution. This construct was also targeted to A-bodies (Fig. 7e, g), where it was insolubilized (Fig. 7h) and immobilized (Fig. 7i) under heat shock conditions. Together, these results suggest that A-body recruitment and aggregation is not based on random protein misfolding, but that specific sensor motifs in unrelated proteins have developed a distinct regulatory role controlling physiological amyloid aggregation.

## Discussion

In this study we explored the mechanisms regulating heat shock-specific protein recruitment and aggregation within A-bodies, using sets of related proteins as a model system. Despite overwhelming similarities in their sequences, predicted structures, and aggregation

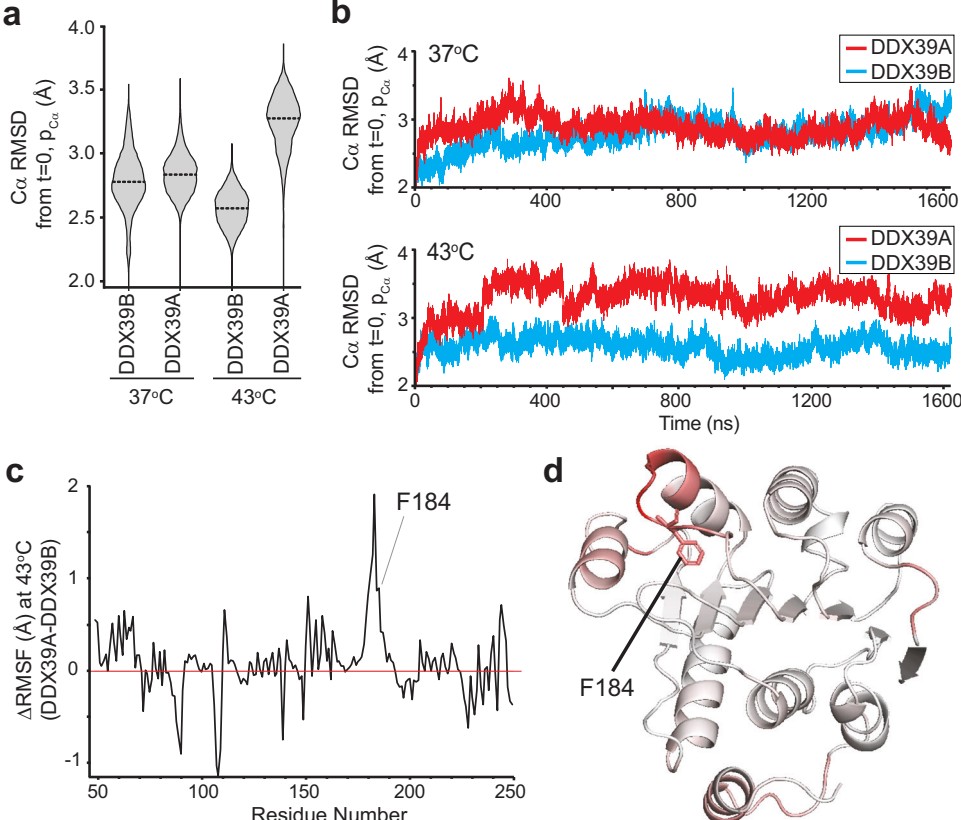

**Fig. 6 | Elevated temperature enhances the dynamic properties of the thermo-sensitive A-body constituent DDX39A. a** Violin plots of root mean square deviation (RMSD) values from 42620 data points across each 1.6 microsecond simulation at 37 °C and 43 °C for the N-terminus (amino acids 46-250) of DDX39A and DDX39B. The dashed horizontal lines in the violin plot represent the mean. **b** RMSD of the N-terminal domain of DDX39A (red) and DDX39B (blue) at 37 °C and 43 °C across a 1.6 microsecond molecular dynamics simulation. **c** The difference in root mean square fluctuations (RMSF) for DDX39A and DDX39B at 43 °C is presented for each residue of the ordered regions of the DDX39 N-terminal domains. **d** RMSF values were converted to hexadecimal color values (red = increased ΔRMSF) and depicted on the DDX39A N-terminal structure. Source data are provided with this paper.

propensity, these proteins are differentially recruited to functional amyloids upon exposure to elevated temperatures. Here, we exploited these similarities to identify discrete structural pockets that control the accessibility of generic A-body targeting sequences. By precisely altering these structural pockets, we were able to modify amyloid aggregation behavior under heat shock conditions, without affecting the trafficking of these proteins under other A-body-inducing stimuli. Thus, we propose a model where the tertiary structure of protein dictates A-body targeting, and acts as an intrinsic sensor to control amyloidogenesis. This effect is likely imparted by a thermo-sensitive conformational shift that occurs in DDX39A and hnRNPA0 (but not the more thermally-stable DDX39B and hnRNPA1), to unmask generic A-body targeting and aggregation domains (Fig. 8).

While there has been scientific progress in the phase transition community, the events mediating the formation of physiological amyloid-like assemblies are not well understood. A-body biogenesis is believed to occur through a multi-step process, as the structure initially possesses liquid-like properties before solidifying as the stress persists[33]. Liquid phase separation often relies on flexible, transient, and multivalent interaction networks between low complexity domains and RNA[10,43]. Similarly, functional amyloid assembly also requires the presence of prion-like or disordered regions in their constituent proteins[27,29,44–47]. An interesting finding in our study demonstrated that although DDX39 and hnRNPA proteins contain intrinsically disordered regions, these low-complexity sequences were not required for imparting stress-specificity or aggregation properties in the A-body pathway. Instead, generic A-body targeting motifs were found in the ordered structures of the DDX39 proteins, buried in

regulatory pockets that were sensitive to temperature changes. Despite the presence of these localization signals within ordered regions of the native structure, we suspect that the appropriate environmental condition could locally denature the regulatory pockets and generate transient low complexity or prion-like regions that facilitate the phase separation events preceding amyloid fibrillation. This notion is supported by molecular dynamics simulations that indicate the N-terminal lobe of DDX39A (but not DDX39B) is sensitive to increased temperatures. This would suggest that A-bodies not only recruit misfolded proteins[48] and unstable peptide fragments[49], but that precise regulatory mechanisms emerged during the structural evolution of A-body constituents to act as intrinsic stress-sensors.

How proteins acquire their native fold is a longstanding and fundamental biological question[50]. However, despite remarkable advances in prediction tools[51,52] and theoretical models[53] the question remains largely unresolved[54,55]. It is fascinating to observe that proteins are generally only marginally stable, with a relatively small energy difference between the folded and unfolded state[56]. DDX39A/DDX39B and hnRNPA0/hnRNPA1 are related protein pairs, which share substantial protein sequence and structural similarities. However, our data demonstrates that their aggregation potential under heat shock conditions is markedly different. We propose that a critical element in the evolutionary divergence of these protein pairs was to tweak the marginal stability of DDX39A and hnRNPA0, and through the addition of thermally-destabilizing mutations tip the protein towards the aggregation state when temperatures increase. Other amino acid substitutions, like those shown not to affect A-body targeting of DDX39 (Fig. 3b), can potentially be associated with the paralog-specific

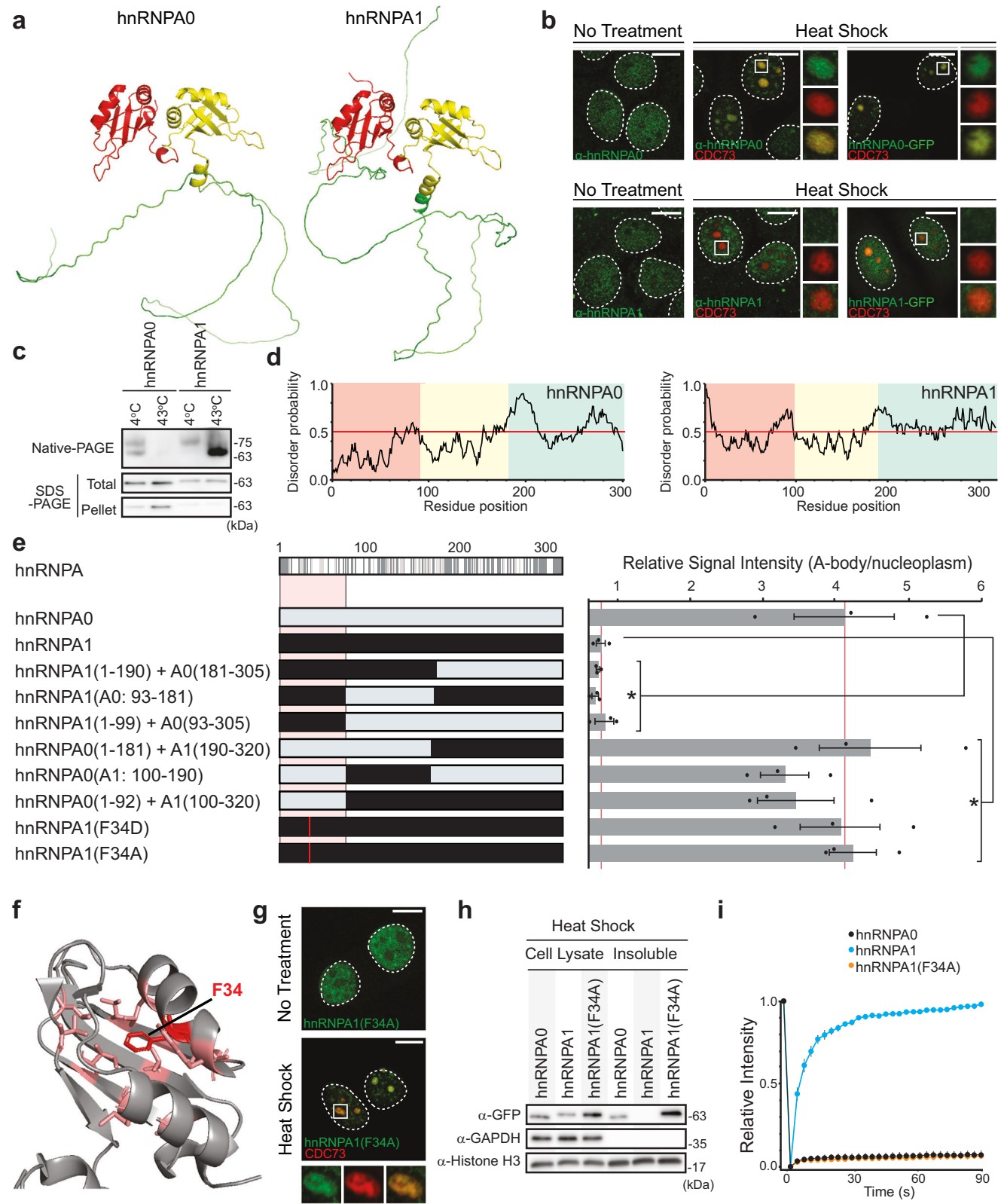

functions. While DDX39B and DDX39A share some redundancy in mRNA processing and export[57–59], the molecular complexes they form allow them to distinguish between different RNA transcripts[36,60]. Similarly, hnRNPA family proteins are involved throughout mRNA maturation, with their altered expression linked to tumorigenesis and neurodegenerative diseases such as ALS[61–63]. By combining functional and structural differences in these RNA interacting proteins, evolution

may have allowed the cell to use marginal stability/selective A-body sequestration to tailor gene expression under heat shock conditions via modulation of RNA maturation.

In general, we believe that this thermo-sensing mechanism can be applied to many additional proteins. Mass spectrometry data indicate there are >250 stress-specific A-body constituents[22], suggesting there could be hundreds of additional thermo-stable and thermo-sensitive

**Fig. 7 | Destabilization of hnRNPA1 regulatory pocket causes heat-induced A-body targeting. a** Cartoon 3D structures of hnRNPA0 (AF-Q13151-F1) and hnRNPA1 (AF-P09651-F1) predicted by AlphaFold[45]. **b** MCF-7 cells expressing the A-body marker CDC73-mCherry were untreated or heat shocked, prior to immunostaining with α-hnRNPA0 or α-hnRNPA1 antibody (green) and imaging. Cells on far right are co-expressing CDC73-mCherry and hnRNPA0-GFP or hnRNPA1-GFP, and were imaged immediately after heat shock. **c** Soluble lysates were extracted from MCF-7 cells expressing hnRNPA0-GFP or hnRNPA1-GFP. Lysates were incubated at 4 °C or 43 °C for 1 h, prior to separation by Native- or SDS-PAGE (Total). Aliquots of the temperature-treated lysates were centrifuged, and pellets were run on SDS-PAGE (Pellet). **d** IUPred3 prediction maps of disordered protein regions in hnRNPA0 (left) and hnRNPA1(right). Background colors correspond to the regions in (**a**). **e** Schematic indicating unique residues (gray lines) within the hnRNPA proteins (top-left). MCF-7 cells expressing the indicated hnRNPA0 or hnRNPA1 chimeric constructs (left) were heat shocked, imaged, and A-body presence was quantified

(right). 10 cells were analyzed per replicate, and values represent means ± s.e.m. (*n* = 3 independent experiments, a two-tailed Student's *t* test was used: *$p \le 0.05$). **f** Cartoon 3D structure of hnRNPA1 (PDB: 1L3K), highlighting the mutation site (F34: red) and surrounding residues that form a hydrophobic pocket (pink). **g** MCF-7 cells expressing the A-body marker CDC73-mCherry and hnRNPA1(F34A)-GFP were left untreated or heat shocked prior to imaging. **h** MCF-7 cells expressing the indicated hnRNPA-GFP constructs were heat shock-treated and lysed. Whole cell lysates and insoluble fractions were extracted, and western blotting detected the hnRNPA (α-GFP), GAPDH (soluble) and Histone H3 (insoluble) proteins. **i** Quantification of FRAP results for heat shock-treated (4 h) MCF-7 cells expressing the indicated proteins. For each quantification 10 cells were analyzed per replicate, and values represent means ± s.e.m (*n* = 3 independent experiments). In representative microscopy images, dashed circles represent nuclei, selected regions (white boxes) are expanded (merge: bottom or far-right), white scale bars represent 10 μm. Source data for all graphs and blots are provided with this paper.

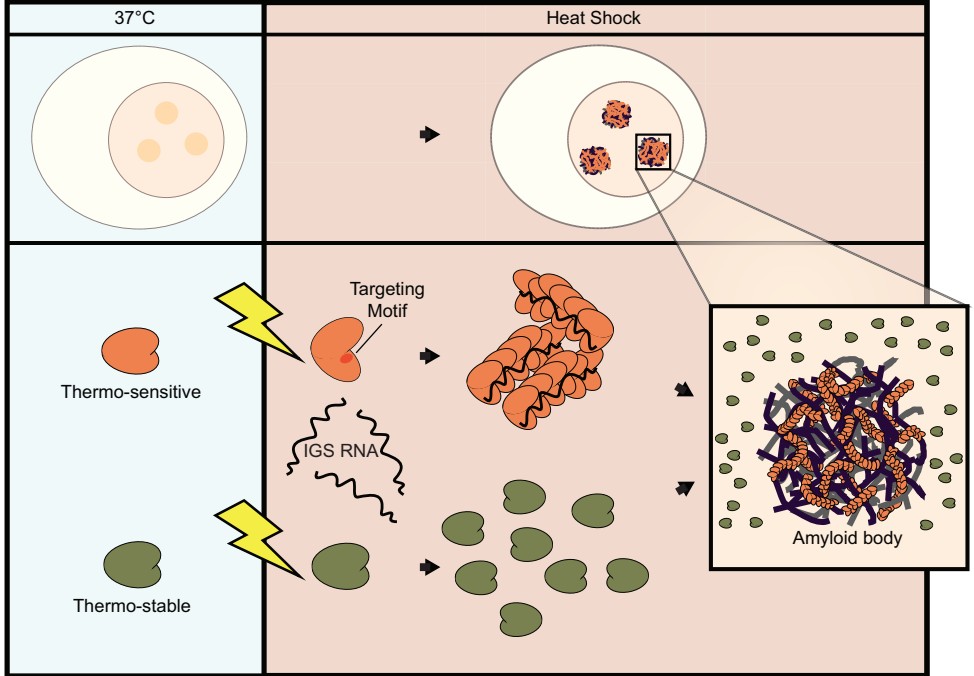

**Fig. 8 | Model depicting the thermo-sensing mechanism of A-body constituents.** Amyloid bodies form in response to elevated temperatures. Under heat shock conditions, thermo-sensitive proteins locally denature to expose targeting motifs that interact with the seeding IGS RNA (intergenic spacer noncoding RNA) and mediate amyloid fibril formation. Conversely, thermo-stable proteins maintain their native conformation at high temperatures, masking their targeting motifs, and preventing protein aggregation.

proteins within this aggregation pathway alone. It is tempting to speculate that a mechanism of stress-sensitive protein stability could also be applied to other A-body-inducing stimuli. It is well established that, like heat, changes in the cellular pH can have a destabilizing effect on proteins[64,65]. Thus, it is conceivable that generic A-body targeting motif could also be masked by structural pockets that are pH-sensitive. Local unfolding under acidotic conditions could be triggered by amino acid R-groups changing from polar to charged (e.g., histidine: $pKa_{(\text{side chain})}$ =6.0), locally denaturing the protein and allowing A-body recruitment and protein aggregation. Evidence from several functional amyloids gives credence to this idea, with *Staphylococcal* biofilm-associated protein containing a pH switch domain that enables formation of amyloid aggregates in acidic environments[66] and the melanosome protein Pmel17 and Neuropeptide Hormone GnRH requiring low pH conditions to fibrillate[67,68]. By extension, proteins containing natively-exposed targeting motifs may represent an additional class of universal A-body targets, with their recruitment and aggregation relying purely on upregulation of the seeding RNA[32,33].

With future studies, we aim to more fully characterize these mechanisms of stress-specific protein aggregation. We envision that this form of intrinsic structural control developed over the course of evolution to be a potent post-translational regulatory mechanism, which doesn't rely on cumbersome and energetically costly covalent modifications. This mechanism likely also extends to other stress response networks, allowing cells to harness the power of protein stability/dynamics for the quick and reversible inactivate cellular pathways in an environmental context dependent manner.

## Methods

### Cell culture, transfection, treatments

MCF-7, A549 and HEK-293 cells (ATCC) were grown in Dulbecco's modified Eagle medium (DMEM) supplemented with 10% FBS and 1% penicillin-streptomycin. Monolayer cultures were grown at 37 °C, 5% $CO_2$ and passaged every 2–3 days. Transfection using PEI (Millipore-Sigma) was performed 1 day following cell seeding, allowing cells to reach ~60% confluency. Unless specified otherwise, heat shock and

acidosis treatments were performed 1 day after transfection for 2 h by placing cells in a 43 °C incubator, or by exchanging growth media with acidic DMEM (pH 6) and placing cells in a humidified Hypoxia chamber at 37 °C, 1%O$_2$, and 5% CO$_2$ (H35 HypOxystation, Don Whitley Scientific).

## Microscopy imaging, immunofluorescence, staining, and quantification

Cells were seeded in 35 mm plates with 20 x 20 mm, 1 mm thick glass coverslips. Following treatment, cells were immediately fixed with cold methanol. Immunostained samples were blocked (10% horse serum) and incubated with α-His antibody (ThermoFisher Scientific, MA1-21315, 1:1000), α-hnRNPA1 (ThermoFisher Scientific, PA5-19431, 1:200) or α-hnRNPA0 (ThermoFisher Scientific, PA5-57722, 1:200) followed by α-mouse Alexa 488 (ThermoFisher Scientific, A11001, 1:400) or α-rabbit Alexa 488 (ThermoFisher Scientific, A11008, 1:400). Thioflavin S (Millipore-Sigma) staining was performed on formaldehyde-fixed, TritonX-100 permeabilized cells[21]. Coverslips mounted in Fluoromount G (ThermoFisher Scientific) were imaged on a Zeiss LSM880 confocal laser scanning microscope with Airyscan (Carl Zeiss Microscopy) using Zen 2.3 software (black edition, Carl Zeiss Microscopy). Image processing was performed with Zen 3.1 (blue edition, Carl Zeiss Microscopy), and no non-linear adjustments were made.

Fixed samples were imaged, and A-body presence was quantified as the average pixel intensity of 1–2 A-body regions relative to the average pixel intensity of the nuclear background for each cell, following image background normalization (Supplementary Fig. 2e). Intensity measurements were generate using ImageJ 1.52a (National Institutes of Health). Values were calculated by averaging 10 images of representative cells in at least three biological replicates. The data is presented as the mean of biological replicates +/- standard error of the mean. Statistical significance was measured via students two tailed t-test, indicated as significant when $p \leq 0.05$.

## Fluorescence recovery after photobleaching

Live cells were grown on glass-bottom 35 mm culture dishes (MatTek Corporation) and visualized by confocal microscopy (Zeiss LSM880 with Airyscan, Carl Zeiss Microscopy). Bleached areas were subjected to a 100% argon laser pulse at 488 nm and imaged for the indicated times post-bleaching. For A-body-detained proteins, a partial region of the A-body was photobleached, while diffuse nucleoplasmic or cytoplasmic proteins had an equal size portion of those regions photobleached instead. Intensity measurements were made using ImageJ 1.52a (National Institutes of Health) and normalized[21,22,31]. All fluorescence recovery after photo- bleaching experiment data represent an average of 10 cells in three independent replicates. The data was normalized[69] and presented as the mean of three biological replicates +/- standard error of the mean.

## Insoluble fractionation, western blotting, and quantification

Treated cells were harvested in NP40 lysis buffer (1% NP40 + 50 mM Tris-HCl + 150 mM NaCl) and fractionated[21]. Briefly, samples underwent two rounds of sonication (60 s, 1 s pulse on/off at 10% power) and centrifugation, prior to re-suspending the insoluble pellet in fresh NP40 lysis buffer. Western blotting used antibodies to detect GFP (Santa Cruz Biotechnology, sc-9996, 1:1000), GAPDH (Santa Cruz Biotechnology, sc-47724, 1:4000), Histone H3 (Cell Signaling Technology, 9715, 1:10,000), hnRNPA1 (ThermoFisher Scientific, PA5-19431, 1:1,000), hnRNPA0 (ThermoFisher Scientific, PA5-57722, 1:1000), and HIS-tag (ThermoFisher Scientific, MA1-21315, 1:5000). α-mouse HRP (ThermoFisher Scientific, A16011, 1:10,000) or α-rabbit HRP (ThermoFisher Scientific, A16023, 1:10,000) were applied prior to detection on the Amerhsam Imager 600 (GE Healthcare Life Sciences). Relative intensities were calculated as the average pixel intensity of heat

shocked insoluble fraction bands compared to the average pixel intensity of the cell lysate bands, normalized against background. Intensity measurements were made using ImageJ 1.52a (National Institutes of Health). For each sample at least three independent blots were measured, and data is presented as the mean value of biological replicates +/- standard error of the mean. Statistical significance was measured via Student's two tailed t-test.

## In vitro aggregation

MCF-7 cells transfected with plasmids encoding DDX39A-GFP, DDX39A(B100-250)-GFP, DDX39B, DDX39B(A100-250)-GFP, hnRNPA0-GFP, or hnRNPA1-GFP were lysed in a non-denaturing buffer (1xPBS + 1 mM PMSF) and sonicated for 60 s (1 s on, 1 s off) to break cell membranes. Insoluble aggregates and other cellular debris were removed from the lysates by centrifugation (5700 x g for 10 min at 4 °C) and total soluble fraction was transferred to a new micro-centrifuge tube. Approximately 100 µg of lysate was incubated at 4 °C and 43 °C for 1 h. Non-denaturing loading buffer (0.01% Brilliant Blue + 10% glycerol + 0.3 M Tris-HCl) was added to native-PAGE samples, while SDS-PAGE samples were boiled for 12 min in denaturing loading buffer (0.02% bromophenol blue +2% SDS + 10% DTT + 10% glycerol + 0.3 M Tris-HCl) prior to loading into 10% TGX-Free FastCast Acrylamide gels (BioRad, 1610173). Native-PAGE electrophoresis was done with a Tris-Glycine (no SDS) buffering system. Aliquots of incubated lysates were also pelleted (12,900 x g for 15 min at 4 °C) and following decanting of the supernatant the pellets were re-suspended in non-denaturing loading buffer, prior to boiling (12 min) and SDS-PAGE. Following electrophoresis, native-PAGE gel was agitated for 15 min in a 0.05% SDS solution, then for 15 min in ddH$_2$O. All samples were transferred to PVDF membrane and immunoblotted with the GFP antibody (Santa Cruz Biotechnology, sc-9996: 1/600).

## Bacterial inclusion body formation and Thioflavin T fibrillation

BL21 cultures expressing GFP, DDX39A-GFP, DDX39B-GFP, or β-amyloid (1-42)-GFP were grown at 32 °C for 20 h. Microscopy was performed on fixed (4% formaldehyde, 1 h) and permeabilized (0.5% Triton X-100, 30 min) bacterial cells, stained with amyloidophilic dye (Congo red) and DNA dye (DAPI).

Thioflavin T fibrillation assays were performed using DDX39A (1-39), DDX39B (1-40), DDX39A (160-199), and DDX39B (161-200) peptides synthesized by GenScript, which were re-suspended to a final concentration of 200 µM in ddH$_2$O[41]. Experiments were carried out in 96-well black-walled plates sealed with optical film. Each well contained a glass bead, Thioflavin T (10 µM), DDX39 peptide (10 µM), and the final volume was adjusted to 100 µL with 1xPBS (pH = 10.5). In a microplate reader (Infinite M200 Pro, Tecan), plates were incubated at 37 °C and shook for one minute between readings (60 s of linear shaking with an amplitude of 1 mm and frequency of 886.9 RPM). Thioflavin T was excited (450 nm) and emission (490 nm) fluorescence was measured every 2 min for 16 h. Data was generated from 3 independent replicates, and presented as the mean value of the independent replicates +/- standard error of the mean. Statistical significance was measured via Student's two tailed $t$ test.

## Plasmids and mutagenesis

Wild-type DDX39A, DDX39B, CDC73, hnRNPA0, and hnRNPA1 cDNA were generated by RT- PCR from MCF-7 RNA and cloned into pEGFP-C1, pmCherry, pET-30, pcDNA3.1-His, pcDNA 3.1/CT-GFP-TOPO (Invitrogen) plasmids. Each of the DDX39A/DDX39B and hnRNPA0/hnRNPA1 domain swapping and point mutations were generated in the pEGFP-C1 or pcDNA 3.1/CT-GFP-TOPO backbones, respectively, using a modified form of site-directed mutagenesis[70]. Construct names refer to the DDX39A or DDX39B, and hnRNPA0 or hnRNPA1 protein backbones with the corresponding amino acid(s) or region(s) substitutions in brackets. Note, DDX39B contains a valine at position 18 that is

absent from DDX39A, leading to a discrepancy when naming equivalent amino acid positions C-terminal of that residue.

## Structural models and protein disorder/aggregation propensity prediction
Structure models were generated, analyzed, and presented using PyMOL Molecular Graphics System (Version 2, Schrödinger, LLC.). Protein structures used in this work were, DDX39B (AF-Q13838-F1 and PDB: 1XTI)[38], DDX39A (generated on a 1XTI template via the Fold and Function Assignment System)[71], the hnRNPA1 RNA-recognition motif (PDB: 1L3K)[72], and full-length hnRNPA1/hnRNPA0 (predicted by AlphaFold)[51]. Disordered protein regions were determined using the IUPred3 interface[73]. Consensus aggregation propensity was calculated using AmylPred 2[42].

## Molecular Dynamics simulation and analysis
DDX39A and DDX39B were first solvated with TIP3P water using a 1.5 nm padding between the edges of the box. NaCl was added to neutralize the system and bring the ionic strength of the solution to 0.15 M. The total number of particles in the resulting systems was ~95,000 atoms (Supplementary Table 1). Simulations are performed with molecular dynamics library OpenMM 7.6 using the Amber99S-BILDN forcefield for protein and TIP3P force field for water with rigid bond constraints and neutralized with counterions. Electrostatics was evaluated with the Particle Mesh Ewald Algorithm using a nonbonded cutoff of 1 nm. A 1.5x hydrogen mass repartitioning was employed, allowing a 4 fs timestep with middle scheme Langevin integrator with a collision rate of 1/ps. The pressure was controlled every 25 steps with a Monte Carlo barostat. The energy was minimized with the steepest descent algorithm, and the system was equilibrated for 50 ps. Each protein was subject to a 1.6-microsecond long simulation at temperatures 37 °C and 43 °C with frames recorded at 40 ps intervals. After processing the solvent out of the resulting trajectories, we separated each system into N- and C-terminal domains and aligned them independently with the starting structure. Using the MDAnalysis python library we calculated the RMSF and RMSD of the C alpha atoms. RMSD was calculated using every 100th frame[74,75]. Violin plots were generated using Prism 10 (GraphPad).

## Statistics and reproducibility
All graphs represent the mean values of at least three biological replicates, with individual data points (dots) overlaid on the bar graph. All sample sizes used in the experiments were in line with those reported in the literature for similar experiments. Error bars represent the standard error of the mean (s.e.m.), and $p$ values were calculated using the two-tailed unpaired Student's $t$ test, with the significance level of $p \leq 0.05$. Representative microscopy and western blot images were captured from experiments that were independently repeated at least three times with similar results.

## Reporting summary
Further information on research design is available in the Nature Portfolio Reporting Summary linked to this article.

## Data availability
The paper and supplementary information contains all the data needed to evaluate the conclusions of the work. Raw data and p values for all Figures and Supplementary Figs. are present in the Source Data files. Initial coordinate, simulation input files, and coordinate files of the final output of the molecular dynamics simulations are available online (https://github.com/PotoyanGroup). DDX39B (AF-Q13838-F1 and PDB: 1XTI, [https://doi.org/10.2210/pdb1XTI/pdb]), hnRNPA0 (AF-Q13151-F1), hnRNPA1 (AF-P09651-F1 and PDB: 1L3K, [https://doi.org/10.2210/pdb1L3K/pdb]) protein structures were used in this work. Any additional information is available upon request to the corresponding author (T.E.A: taudas@sfu.ca). Source data are provided with this paper.

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

## Acknowledgements

This work was supported by the Canadian Institute of Health Research (T.E.A., PJT-162364) and Natural Sciences and Engineering Research (T.E.A., RGPIN/04998-2017). D.A.P. acknowledges funding from the National Institute of General Medical Sciences (NIGMS) of the National Institutes of Health for financial support (R35 GM138243). T.E.A. acknowledges the kind support of the Canada Research Chairs program for a Tier II Canada Research Chair: Cellular Stress (CRC-2021-00117).

## Author contributions

DM performed experiments, with assistance from EAM, SC and RZ. DM, HW, DAP and TEA designed the experiments. DB and DAP performed molecular dynamics simulations and analyses. DM and TEA conceptualized the study and wrote the manuscript. All authors contributed to review and editing. TEA supervised the project.

## Competing interests

The authors declare no competing interests.
