## [Peer Review File · Nature Communications]

REVIEWER COMMENTS

Reviewer #1 (Remarks to the Author):

In this manuscript, the authors study the ways cells react to stimuli by forming amyloid bodies, which are RNA-seeded nuclear condensates. They study two closely related RNA helicases, DDX39A, which is targeted to both acidotic and HS induced A-bodies, and DDX39B, which is only targeted to acidotic A-bodies. They find that by manipulating the structural pockets of these proteins, targeting to A-bodies can be modulated. Interestingly, they find that the aggregation of proteins within A-bodies is dictated by thermal stability. This suggests a model whereby temperature sensitive structural regions regulate the processing of proteins, allowing for rapid response to stress. The investigation is quite comprehensive with respect to sequence/structure mediating the differences between DDX39A and B. The authors do a nice job of comprehensively screening the helicases via mutagenesis to understand the mechanism of targeting. I think this is a nice study that has the potential to be of broad interest to the research community. However, I believe that there are also aspects that are somewhat incomplete. I also have several suggestions for improvement of the data analysis and presentation.

Major comments:

- My biggest concern is that while the experiments go into great depth with exploring the effects of numerous mutations on DDX39 structure, the fundamental question of the physiological relevance of these mutations is not explored. In this way, the study seems incomplete. Is there a physiological/pathological effect? What proteins are/are not targeted to the A bodies due to these mutations? How do these mutations modulate helicase activity? I suggest selecting a small number of mutations and validating their physiological relevance.

- Along these same lines, the authors show that heat induced targeting of DDX39 and hnRNP is facilitated by portions of the globular region of the proteins rather than the IDR regions. What is the effect of F184, C223, and the F34 mutations on the physiological activity of DDX39 helicases and hnRNP function?

- Figure 1- there is no quantitation in this figure. In place of quantitation, they state that they have shown similar results in multiple cell lines (Fig S1C). However, this supplemental figure only shows colocalization of 39A with 39B, not colocalization with CDC73 and/or ThS. The authors should either quantify Figure 1A (preferable) or add co-staining to the supplemental figure.

- For the FRAP data, in nearly all of the panels it appears that the bleaching in each curve was inconsistent (in Fig 1E the red curve was only bleached to ~40% while the control curves are bleached to ~15%). This may be due to rapid recovery already occurring and competing with bleaching, but in

Supplementary Figure S1 the blue curves are only bleached to ~50% and do not recover as quickly as the DDX39B/HS curve. This is also very noticeable in Figure S5. Is there an explanation for this?

- Also, in Fig 1E and S1B, and throughout the paper, the legends state that the plots show quantitation of 7 cells. However, there are no error bars and it is unclear if this data represents multiple trials. Based on the methods section, it appears that these are averages, but error bars should be shown. Also representative images should be shown to depict bleach and post bleach recovery.

- Figure 3 nicely shows the trends in targeting by the different DDX constructs under heat shock conditions. Figure 3C is meant to be a control showing that the same substitutions have no effect on sequestration in acidosis induced A bodies. However, in Figure 3C single cells are shown with no quantitation. It is useful to show these images, but quantitation should also be included to have a better sense of the variability when inducing via heat shock vs acidosis.

- The authors show that DDX39A is ThS positive. More discussion could be added regarding if the thought is that DDX39A is an amyloid protein on its own like α -synuclein and tau, or if the A bodies are thought to just contain amyloid-positive proteins within.

Minor comments:

- In Figure 3E the labeling should be checked. Should this be DDX39B (L185F+V224C)?

- In Figure 4G the labeling appears incorrect and this mutation should be T114X and not T144X.

- For the structures in Figures 4 and 5, coloring by atom (nitrogen=blue, oxygen=red) should also be used for each of the amino acids of interest. This will assist in viewing the character of the structural regions that are described.

- In Figure 6 the abbreviations have typos. I think this should be RMSD not RSMD (panel A and B) and RMSF and not RSMF (panel C)

- In results section 5, line 239, 241, and 242 I think the figure callout should be FS6E and not S6D.

Reviewer #2 (Remarks to the Author):

Understanding the changes in the form and function of biomolecules in response to environmental stimuli is one of the key goals in biology. In this study, Marijan et. al. investigated the molecular

mechanisms for the heat shock protein sequestration into amyloid bodies (A-body). Their previous proteomic study identified two closely related RNA helicases with 90% sequence identity, DDX39A and DDX39B, possessing different responding mechanisms to heat shock. Here, the authors confirmed that DDX39A localized in the A-body as insoluble and immobile, while DDX39B remained nuclear, soluble, and mobile under heat shock. Using a series of substitution constructs, truncations, and single/combined mutations, the authors characterized the structural features that mediate heat shock-specific sequestration.

Though it is an extensive characterization, there are several concerns:

- 1) It is still unclear why DDX39A can be recruited in A body while DDX39B can not under heat shock conditions. Experiments using substitution constructs indicate that residues 100-250 determine the heat shock-specific sequestration in the full-length molecule. However, truncation experiments showed that the truncated fragment 100-250 does not capture the heat shock-specific sequestration phenotype. Only combining residues 50-75 with 100-250 can prevent the aggregation of DDX39B under heat shock. The following mutation experiments mostly characterized the structural differences in the region of 100-250 between DDX39A and DDX39B. However, how the differences in 100-250 work with 50-75 to cooperatively determine the heat shock-specific sequestration remains unknown.
- 2) Based on Fig. 2C, the next 25 residues after the first ~50 amino acids are found within a high-order region in both DDX39A and DDX39B. This result does not provide evidence for the different responses of DDX39A and DDX39B to heat shock. Descriptions in lines 111-114 do not support descriptions in lines 114 – 118.
- 3) For the molecular dynamic simulation in Fig.6C, besides the difference observed around I153, there is also a big difference for residue 60~70. Does it mean some long-range associations between residues around I153 with residues 60~70? The manuscript does not describe the strike difference for MD simulation for residues 60~70, which seems important to prevent the aggregation of DDX39B.
- 4) The authors suggest that thermal stability determines the heat shock-specific sequestration. Can the authors characterize the thermal stability of DDX39A and DDX39B (or relative mutant/chimera)? MD simulation of DDX39A is based on a predicted structure, while DDX39B's modeling is based on a crystal structure. It would be more convincing if the authors could verify the changes in protein thermal stability using purified protein in experiments.
- 5) Line 199, what is the residue numbering for the N-term lobe?
- 6) Amyloid fibril is structurally dominated by β -sheet structure. What proteins construct the β -sheet amyloid fibril in the amyloid-like body? For example, do amyloidogenic sequences predicated by AmylPred2 in DDX39 form amyloid fibril in vitro?

Reviewer #3 (Remarks to the Author):

Amyloid bodies (A-bodies), an RNA-dependent subnuclear condensate, are an example of functional amyloid which forms due to the cellular stress response. However, little is known regarding the mechanism that underlies the production of physiological amyloid assemblies. In this manuscript, the authors have proposed a model where the tertiary structure of the proteins, along with their thermal stability, collectively dictates their selective aggregation within amyloid bodies (A-bodies). Here, the authors have made a library of mutations in A-body-related proteins (DDX39 and hnRNPA) to probe the mechanism that regulates heat shock-mediated protein recruitment within the A-bodies and their further aggregation. Although the set of related proteins has enormous similarities in their predicted structures, sequences, and propensity to form aggregates, however, they show differential recruitment inside A-bodies when exposed to increased temperature. Based on their rigorous mutation and A-bodies targeting study, the authors have determined a crucial structural element that regulates heat-shock-specific amyloid aggregation, which can either be induced or restricted by manipulating their structural pockets. The study also concludes that the intrinsically disordered regions did not contribute to the A-bodies targeting property, which is a fascinating outcome of the manuscript.

The experiments were well thought out and executed elaborately with adequate discussion. The manuscript mainly focuses on preparing a mutation library and A-bodies targeting study, which is considerable work. However, the paper lacks a discussion on the in-vitro characterization of A-bodies formed in different conditions. Moreover, the microscopy images throughout the manuscript are not up to the journal's standard. I have the following questions.

1. The introduction does not explain the role of two sets of model proteins used in this manuscript, DDX39 and hnRNPA. The authors should briefly describe these two proteins highlighting their relevance and functionality for A-body formation.
2. In the introduction section, at the end, the author should briefly discuss their main finding and conclusion. The introduction section looks abruptly ends without any conclusion.
3. The author has mentioned in line 65 (also illustrated in Figure 1A) that DDX39A and DDX39B show 90% amino acid sequence identity. The authors should indicate the sequence homology with sequence alignment study and incorporate that in the supplementary information.
4. In Figure 1B, MCF-7 cells have been transfected with DDX39A-GFP, DDX39B-GFP, and the amyloid body marker protein CDC73-mCherry. Further, cells were left untreated or exposed to heat shock or extracellular acidosis. Though the cells treated with heat shock were stained with Thioflavin S (ThS) dye, the cells treated with extracellular acidosis were not stained with ThS dye. The authors should also show the ThS result for the acidosis study. Moreover, this figure does not clearly show the colocalization between the A-bodies marker and DDX39 proteins. Authors should provide better images for colocalization and can also calculate the percentage of colocalization based on their intensity profile. Another question that strikes in this figure is as follows. The cells were exposed to heat shock for 4 hrs at 43 °C. Is there any reason or reference for that specific temperature and time duration utilized for heat

shock? If yes, the authors should clarify that. Otherwise, the author should show and incorporate the optimization study in the supplementary section.

5. Figure 1C shows Western blots for the range of substitution constructs of DDX39 expressed in MCF-7 cells in untreated or heat-shock-treated cells. Here, the blots lack loading control for both the cell lysate and insoluble fraction. The author should show the gel image with loading controls GAPDH and Histone H3 for soluble and insoluble fractions in the supplementary section for a fair comparison.

6. Figures 1E, 7H, S2 B, S3 A and S3 C, S5 C Quantification of FRAP Data have been illustrated without normalization. The FRAP data can better be represented as normalized intensity (with positive and negative background correction) and data fitting for better comparison.

7. In Figure 2A, A-body targeting efficiency has been calculated as the average A-body pixel intensity relative to the average nuclear background pixel intensity. Since a significant amount of work is based on the quantitative microscopy approach, the authors should elaborate on the calculation from the microscopy for at least one representative data set in the supplementary section for a better understanding of the calculation.

8. In line 98, the authors have demonstrated via a quantitative microscopy approach that GFP is sequestered by the amino acids 75-250 proteins when exposed to heat shock conditions. In this case, the author should confirm the statement by utilizing an A-body marker and colocalization experiment to confirm whether the GFP-tagged proteins co-localize with the A-body marker.

9. Figure 2C: IUPred3 prediction maps of disordered protein regions for DDX39A and DDX39B proteins have been shown. The authors should also mention the amino acid domains and the IUPred prediction graph, which will provide domain-specific structural information.

10. Figure 3B and 3D quantify the A-bodies targeting efficiency of point mutations that replace each of the unique residues in DDX39A with their DDX39B equivalents and vice versa. However, the amino acid positions in Figure 3B and the corresponding reciprocal mutations in Figure 3D are different by one unit throughout all the variants (for example, F184L in Figure 3B and L185F in Figure 3D). Am I missing something in this figure? The author should recheck the designated position of point mutations and clarify the concern.

11. In Figure 3B and 3D, the authors have mentioned that “the substitution of F184L or C223V reduced A-body localization and increased mobility of the mutant DDX39A proteins to levels similar to that of the DDX39B” thus exerting a “masking effect” on DDX39A. If these two single-point mutations pose the masking effect on the protein, then it is expected that during the reciprocal substitution where amino acids L from DDX39B is substituted to amino acid F from DDX39A would release the masking effect with an enhancement in the A-bodies targeting efficiency. But the representative data in Figure 3D shows the opposite of the expected data. The author should clarify the apparent discrepancy in the representative data compared to that of the expected data.

12. In Figure 4D, the Author should show the individual effects of the three mutations (I106L+N188H+V167I) other than the T114S mutation as well to precisely depict the effect of individual mutation on the relative signal intensity from A-body.

13. In Line 195, the authors have investigated the root mean square deviation (RMSD) calculation of DDX39A by using 52 °C as high temperature and 27 °C as low temperature. The authors should clarify the basis for choosing these specific temperatures.

14. Figure 7F shows heat shock-treated MCF-7 cells expressing the A-bodies marker CDC73-mCherry and hnRNPA1 (F34A)-GFP. Even though the insets display the CDC73-mCherry signal, the authors should provide a merged image for a better demonstration of the colocalization.

15. In Figures 7G, S1 A, and E, in both the Western blots, it is quite evident from the blot that the GAPDH expression is inconsistent throughout the lane (more inconsistent in S1 E). In this case, the band's intensity should be calculated, analyzed, and compared.

16. Figure S2 C shows heat-shock-treated MCF-7, A549, and HEK293 cells expressing DDX39A-GFP and DDX39B-mCherry. This experiment illustrated that the cell line is not responsible for mediating the divergence in stress-specific amyloid body targeting. However, a question that arises here is why the cells were only heat-shock treated and not exposed to acidosis. The author should clarify the point.

17. Figure S6: Please review the figure, as the figure legends do not match the illustrated data.

18. The authors should perform immunostaining with amyloid-specific OC antibody to confirm the amyloid state of the A-bodies formed in cells when exposed to heat shock and acidosis. Moreover, the authors should also perform electron microscopy of the A-bodies isolated from cells to visualize their morphology. Apart from Thio-S staining, I don't see any effort from the author to characterize the A-body for the amyloid formation,

19. Apart from the in-cell studies performed in the manuscript, which are indeed rigorously done, the authors should also perform in-vitro studies to characterize the A-body aggregates through biophysical experimentation (e.g., CD, Thioflavin T assay, etc.). I would encourage the author to do in vitro experiments with DDX39A and B and a few mutants for their thermal response to the structural changes (secondary or tertiary structure) and aggregation propensity (using ThT fluorescence, CD, and electron microscope). That might correlate with the cellular data, establishing the structure-function(A-body formation here).

Reviewer #1:

In this manuscript, the authors study the ways cells react to stimuli by forming amyloid bodies, which are RNA-seeded nuclear condensates. They study two closely related RNA helicases, DDX39A, which is targeted to both acidotic and HS induced A-bodies, and DDX39B, which is only targeted to acidotic A-bodies. They find that by manipulating the structural pockets of these proteins, targeting to A-bodies can be modulated. Interestingly, they find that the aggregation of proteins within A-bodies is dictated by thermal stability. This suggests a model whereby temperature sensitive structural regions regulate the processing of proteins, allowing for rapid response to stress. The investigation is quite comprehensive with respect to sequence/structure mediating the differences between DDX39A and B. The authors do a nice job of comprehensively screening the helicases via mutagenesis to understand the mechanism of targeting. I think this is a nice study that has the potential to be of broad interest to the research community. However, I believe that there are also aspects that are somewhat incomplete. I also have several suggestions for improvement of the data analysis and presentation.

We appreciate the effort that the Reviewer has put into providing us with a thorough review of our manuscript, and are encouraged by their kind words regarding the broad interest this work will have within the research community. We have made a strong effort to improve the analysis and presentation of the data (including the addition of 21 new and 18 revised figure panels), and believe that the Reviewer's comments have allowed us to substantially improve the quality of our manuscript. Regarding the incompleteness that the Reviewer felt about aspects of the story, we absolutely acknowledge that much still needs to be learned about A-bodies. As you will see in our responses below, there are technical and conceptual limitations that prevent us from moving in some of the directions suggested at this time. However, we firmly believe that this study provides comprehensive mechanistic insight into an emerging area of structural, molecular, and cellular biology (e.g., phase separation and amyloid aggregation).

Major comments:

1) My biggest concern is that while the experiments go into great depth with exploring the effects of numerous mutations on DDX39 structure, the fundamental question of the physiological relevance of these mutations is not explored. In this way, the study seems incomplete. Is there a physiological/pathological effect? What proteins are/are not targeted to the A bodies due to these mutations? How do these mutations modulate helicase activity? I suggest selecting a small number of mutations and validating their physiological relevance.

This manuscript identifies a novel thermo-sensing mechanism that uses the tertiary structure of proteins to drive stress-specific condensation/aggregation. Furthermore, our results showing that high-order regions of target proteins mediate this phenomenon is divergent from the prevalent belief that intrinsically disordered regions are the key regulatory domains, a point noted by Reviewer 3, "*The study also concludes that the intrinsically disordered regions did not contribute to the A-bodies targeting property, which is a fascinating outcome of the manuscript.*" Thus, the physiological relevance/effect of the amino acid substitutions in this paper is a shift in the stress-specific aggregation profile of the proteins, and based on the knowledge we gained, we were able to modify a protein's stress-specific aggregation profile by introducing a single amino acid substitution. Overall, we believe this represents a significant conceptual advancement in our understanding of this aggregation process.

Regarding the next two questions in this comment, we share the Reviewer's intrigue and desire to move our understanding of A-body biology forward. However, the scope of this work must be managed, and we put forward that the later questions have conceptual hurdles that prevent us from pursuing those directions at the moment (discussed below).

What proteins are/are not targeted to the A bodies due to these mutations?

Upon completing our analysis of DDX39, we sought to generalize our study by considering a second pair of proteins (hnRNPA0/hnRNPA1). We found that these RNA splicing factors also use ordered structural regions

to control heat shock-specific aggregation, and using the knowledge we gained from DDX39 we could precisely manipulate the stress-specific aggregation of hnRNPA0 as well.

Generating a complete list of proteins that utilize this mechanism is not a trivial task, and we are unsure whether it would advance our mechanistic understanding of this pathway. Our MS data has found that there are hundreds of cellular proteins with stress-specific targeting profiles (Marijan, 2019, *FEBS Letters*), suggesting that this pathway could be broadly utilized to control protein aggregation. However, identifying and validating additional candidates would require a comprehensive analysis of the structures of each protein, and the testing of potentially hundreds of structural features to assess their thermo-sensitivity. We hope the Reviewer will agree that this goes beyond the scope of our work.

How do these mutations modulate helicase activity?

When we conceived this study, our goal was to uncover the mechanism regulating stress-specific amyloid aggregation. The only way we could tackle this problem was to select pairs of closely related proteins, with Protein X recruited to heat shock-induced A-bodies and a the closely related Protein Y not targeted. This strategy allowed us to create a huge panel of substitution constructs that were necessary to map the regulatory elements described in our manuscript. However, an unfortunate consequence of this rigid selection criteria was that it forced us to choose proteins that may or may not have established functional assays.

In fact, DDX39A (50 PubMed citations) and hnRNPA0 (26 PubMed citations) are extremely poorly characterized in the literature, with most studies on DDX39A loosely tied to clinical effects (e.g., anti-viral infection and cancer metastasis). The only paper with a clear functional readout for DDX39A, links this RNA helicase to mRNA nuclear export. However, this is not useful in our setting, as it is well-established that nuclear/cytoplasmic transport is suppressed under heat stress conditions (Saavedra, *et al.*, 1996, *Genes & Development* and Tani, *et al.*, 2017, *MBoC*). As cells encode dozens of RNA helicases and splicing factors, we are at a loss to parse out the roles of these proteins without further growth in the DDX39A/hnRNPA0 literature. We hope the Reviewer will agree that mapping the currently undefined roles of DDX39A and hnRNPA0 are beyond the scope of this study and appreciate our limitations in this area.

2) Along these same lines, the authors show that heat induced targeting of DDX39 and hnRNP is facilitated by portions of the globular region of the proteins rather than the IDR regions. What is the effect of F184, C223, and the F34 mutations on the physiological activity of DDX39 helicases and hnRNP function?

As discussed above, DDX39A and hnRNPA0 are poorly characterized and there are no viable functional assays to explore the effects of the amino acid substitutions on these proteins. It is also worth noting that functional assays using the better characterized DDX39B and hnRNPA1 would not be relevant here, as these proteins are not targeted to A-bodies under heat shock conditions, leaving their physiological activity putatively unaffected at elevated temperatures.

Conceptually, we hypothesize that the amino acid differences between DDX39A and DDX39B (45 differences in 428 residues) could be broken into two categories. Residue changes that; (1) effect stress-specific aggregation (i.e., those that reduce A-body targeting in Figure 3) and (2) control the functional differences of the closely related helicases. As many of the substitutions used in this manuscript change a DDX39A amino acid to the associated DDX39B residue (or vice versa), it would be logical to suppose that the substitutions that don't change aggregation dynamics would push the functionality of DDX39A towards DDX39B. We have no assays to explore this hypothesis further, but like the Reviewer we are very interested in this area and hope the literature builds to a point that we can test this in the future.

3) Figure 1- there is no quantitation in this figure. In place of quantitation, they state that they have shown similar results in multiple cell lines (Fig S1C). However, this supplemental figure only shows co-localization of 39A with 39B, not co-localization with CDC73 and/or ThS. The authors should either quantify Figure 1A (preferable) or add co-staining to the supplemental figure.

We appreciate the Reviewer's feedback and have implemented both suggestions, adding quantification to the data in Figure 1 (**new Figure 1D, S4B**) and CDC73 co-localization to the cell line data originally in Figure S1C (**new Figure S2A,B**)

In Figure 1, quantification of the A-body targeting efficiency of DDX39A and DDX39B under no treatment, acidosis, and heat shock conditions was added as a **new Figure S4B** to aid in the analysis of the F184L and C223V substitutions (see comment 6, below). We also shifted the western quantification of the stress-specificity domain mapping to **Figure S2F** and performed a new microscopy quantification of A-body targeting for all of the mutants in **Figure 1D** (the original representative images of the critical domain substitutions are still included in Figure 1E). Also, to aid the Reader in understanding the microscopy quantification methodology, we added a panel describing the calculations used to generate Relative Signal Intensity (**new Figure S2E**).

In **Figure S2A and S2B** we added co-staining with CDC73, to confirm the co-localization of DDX39A/DDX39B with an A-body marker in the A549 and HEK293 cell lines

4) For the FRAP data, in nearly all the panels it appears that the bleaching in each curve was inconsistent (in Fig 1E the red curve was only bleached to ~40% while the control curves are bleached to ~15%). This may be due to rapid recovery already occurring and competing with bleaching, but in Supplementary Figure S1 the blue curves are only bleached to ~50% and do not recover as quickly as the DDX39B/HS curve. This is also very noticeable in Figure S5. Is there an explanation for this?

The Reviewer aptly notes that recovery dynamics can be heavily dependent on the recovery rate of the protein and the stress state of the cell. With this in mind, we shifted to a normalized FRAP recovery calculation (recommended by Reviewer 3), where the intensity of the first post-bleach image is set to zero.

This standardizes all the proteins and treatment conditions, making the relative assessment of data much simpler. This change was applied to every FRAP figure (Figure 1F, 7I, S1C, S1D, S4A, S6C, and S7D), and a representative quantification is displayed to the right (Figure 1F, which replaced Figure 1E).

5) Also, in Fig 1E and S1B, and throughout the paper, the legends state that the plots show quantitation of 7 cells. However, there are no error bars and it is unclear if this data represents multiple trials. Based on the methods section, it appears that these are averages, but error bars should be shown. Also representative images should be shown to depict bleach and post bleach recovery.

During the revisions process, we reproduced all the FRAP data (Figure 1F, 7I, S1C, S1D, S4A, S6C, and S7D) to generate 3 independent replicates with 10 cells/replicate. The data is presented as the mean of the three replicates, with error bars (standard error of the mean) included for all samples. It should be noted that in some cases the error bars are so small that they are covered by the data points.

We have also included representative images for the first FRAP samples described in the manuscript (Figure S1C and S1D). In our experience, it's uncommon to see representative images for all FRAP experiments, and due to space limitations, we were not able to include them in our manuscript. However, we appreciate the Reviewer's recommendation, and feel that including these representative images in Figure S1C/D will help the Reader gain important context on the FRAP experiments done throughout the manuscript.

6) Figure 3 nicely shows the trends in targeting by the different DDX constructs under heat shock conditions. Figure 3C is meant to be a control showing that the same substitutions have no effect on sequestration in acidosis induced A bodies. However, in Figure 3C single cells are shown with no quantitation. It is useful to show these images, but quantitation should also be included to have a better sense of the variability when inducing via heat shock vs acidosis.

As the Reviewer requested, we have quantified the A-body targeting efficiency of DDX39A, DDX39B, DDX39A (F184L), and DDX39A (C223V) under no treatment, heat shock, and acidotic conditions.

As can be seen in the graph (new Figure S4B), all four proteins show a similar level of sequestration under acidotic conditions, indicating that the F184L and C223V substitutions do not affect acidotic A-body targeting.

7) The authors show that DDX39A is ThS positive. More discussion could be added regarding if the thought is that DDX39A is an amyloid protein on its own like a-synuclein and tau, or if the A bodies are thought to just contain amyloid-positive proteins within.

Considering the conformational state of individual A-body constituents is always a challenging question. Based on transmission electron microscopy results and Congo red- or Thioflavin S-positive staining (Audas, *et al.*, 2016, *Dev. Cell*) there is clearly a large population of amyloid-like proteins in the A-bodies.

When we consider whether an individual protein adopts an amyloid conformation, we look for a shift towards insolubility and immobility, when the protein enters an A-body. These properties would align with proteins that have fibrillated and would be counter-indicative of mobile/transient residents.

To take our analysis a step further, we assessed whether DDX39A and DDX39B have the capacity to convert to an amyloid conformation, as this would markedly increase our confidence that they adopted an amyloid-like state upon entering A-bodies. Here, we expressed the DDX39 proteins in bacteria and found that they could form inclusion bodies (a known amyloid model) (new Figure 1C), and our DDX39 peptides were capable of fibrillation in a Thioflavin T aggregation assay (new Figure 2C, S3D). Together, these data suggest that DDX39A/DDX39B can adopt an amyloid conformation and increase our confidence that they do so within the A-bodies. We also included text discussing this in the manuscript:

Line 83: “For these RNA helicases, the putative adoption of an amyloid-like conformation within A-bodies was demonstrated by the hallmark shift towards the insoluble (Figure S1B) and immobile (Figure S1C-D) biophysical state and the inherent capacity of these proteins to generate fibrils, when expressed in a bacterial inclusion body assay (Figure 1C: a model setting of amyloid aggregation^{21,39,40}).”

Line 132: “As AmylPred2⁴² predicts that each of these regions contain aggregation prone clusters (Figure S3B-C – Bottom Panel), we synthesized DDX39 peptides and ran a Thioflavin T (ThT) fibrillation assay to determine whether DDX39 fragments could adopt an amyloid conformation. In this assay, peptides from the central region of DDX39A (160-199) and DDX39B (161-200) generated ThT-positive amyloid fibrils at a significantly higher rate than AmylPred2-negative DDX39A (1-39) and DDX39B (1-40) regions (Figure 2C, S3D).”

Minor comments:

- In Figure 3E the labeling should be checked. Should this be DDX39B (L185F+V224C)?

Thank you for catching this, the figure labels have been corrected.

- In Figure 4G the labeling appears incorrect and this mutation should be T114X and not T144X.

Thank you for catching this too, this figure labels have also been corrected.

- For the structures in Figures 4 and 5, coloring by atom (nitrogen=blue, oxygen=red) should also be used for each of the amino acids of interest. This will assist in viewing the character of the structural regions that are described.

We appreciate the Reviewer's feedback. However, we believe that the color-coding used in Figure 4 and 5 is an important tool to help the Readers follow the many mutations introduced throughout the manuscript.

Additionally, we generated a figure (right) where side chain atoms (not backbone atoms) within the two hydrophobic pockets were colored blue (nitrogen), red (oxygen), and yellow (sulphur). Since there are only three weakly polar sulphur residues (two methionine and one cysteine) and no nitrogen or oxygen atoms within the two pockets we didn't feel including this image would substantially improve the Readers understanding of the work. We hope the Reviewer can understand our position on this point.

- In Figure 6 the abbreviations have typos. I think this should be RMSD not RSMD (panel A and B) and RMSF and not RSMF (panel C)

Our apologies for missing this. The labels have been changed.

- In results section 5, line 239, 241, and 242 I think the figure callout should be FS6E and not S6D.

We have attempted to correct all miss-labeled callouts throughout the manuscript.

Reviewer #2:

Understanding the changes in the form and function of biomolecules in response to environmental stimuli is one of the key goals in biology. In this study, Marijan et. al. investigated the molecular mechanisms for the heat shock protein sequestration into amyloid bodies (A-body). Their previous proteomic study identified two closely related RNA helicases with 90% sequence identity, DDX39A and DDX39B, possessing different responding mechanisms to heat shock. Here, the authors confirmed that DDX39A localized in the A-body as insoluble and immobile, while DDX39B remained nuclear, soluble, and mobile under heat shock. Using a series of substitution constructs, truncations, and single/combined mutations, the authors characterized the structural features that mediate heat shock-specific sequestration.

Though it is an extensive characterization, there are several concerns:

We would like to thank the Reviewer for their thoughtful consideration of our work, and we wholeheartedly agree that understanding how biomolecules adapt to changing environmental conditions is a key goal in cell biology. We have responded to each of the points raised by the Reviewer (below), and we hope they will see that their feedback has allowed us to significantly improve the quality of our manuscript.

1) It is still unclear why DDX39A can be recruited in A body while DDX39B can not under heat shock conditions. Experiments using substitution constructs indicate that residues 100-250 determine the heat shock-specific sequestration in the full-length molecule. However, truncation experiments showed that the truncated fragment 100-250 does not capture the heat shock-specific sequestration phenotype. Only combining residues 50-75 with 100-250 can prevent the aggregation of DDX39B under heat shock. The following mutation experiments mostly characterized the structural differences in the region of 100-250 between DDX39A and DDX39B. However, how the differences in 100-250 work with 50-75 to cooperatively determine the heat shock-specific sequestration remains unknown.

Our model suggests that proteins possess generic A-body targeting and aggregation motifs (Figure 2), which can be buried within the tertiary structure of a protein (Figure 4-5). Heat shock exposure causes perturbations in the structure of thermo-sensitive (DDX39A and hnRNPA0), but not thermo-stable (DDX39B and hnRNPA1) proteins, making these buried motifs accessible (in DDX39A/hnRNPA0) and leading to A-body aggregation. Specific residues (e.g., F184, C223, and T114 in DDX39A) are critical for imparting thermo-sensitivity (Figure 3), perhaps acting as keystones in structural pockets that cover the aggregation motif. Notably, molecular dynamics simulation (Figure 6) and *in vitro* data (Figure 1G and 7C) validated this model, by highlighting the thermo-sensitivity of DDX39A and thermo-stability of DDX39B. Together, this makes a protein's tertiary structure a direct thermo-sensor within a cell, deciding the A-body targeting fate for individual molecules.

When considering Figure 2, we have added new text and data to clarify the role of amino acids 50-74. While we started with two potential hypotheses (below), we believe the data clearly excludes the former:

- (1) This region could contain a discrete motif (e.g., protein or RNA binding site) that cooperates with residues 100-250 to impart stress-specific targeting.
- (2) This region could be critical for protein folding, as adoption and maintenance of the native fold is necessary to impair heat shock-specific sequestration.

Hypothesis 1: Based on our data, it seemed unlikely that amino acids 50-74 contains a discrete motif that actively inhibits A-body targeting, as it; (1) falls outside of the stress-specificity region identified in **Figure 1**, (2) contains no amino acid substitutions unique to DDX39B (**Figure S1A**), and (3) fails to disrupt A-body targeting in DDX39B (1-150)/DDX39B (1-200) (**Figure 2A**). During revisions we

also made a new construct that directly fusing the 50-74 region to DDX39B (100-250), called DDX39B (50-74+100-250). Here, we reason that if a motif in the 50-74 region can cooperate with amino acids 100-250 to impart stress-specificity, this hybrid constructs would possess the phenotype of the full-length proteins. As can be seen from the data (**new Figure S3E**), the DDX39B (50-74+100-250) does not possess stress-specificity and is targeted to A-bodies under heat shock conditions. Together, these data distanced us from the co-operative motif hypothesis.

Hypothesis 2: The AlphaFold structure (AF-Q13838-F1) and an IUPred3 analysis predict that the first 50 residues of DDX39B are intrinsically disordered, while the next 25 amino acids are in the ordered structure. This would support the notion that DDX39B (50-250) can adopt a native conformation, but removing the next 25 high-order residues (DDX39B [75-250]) would have a more profound effect on protein folding. Considering the data disputing hypothesis 1, we took this as a push towards the second hypothesis and used this figure as a “jumping-off” point. Our work in Figures 4-7 strongly support the concept that tertiary structure and protein folding are the critical determinants of stress-specific A-body targeting and aggregation.

2) Based on Fig. 2C, the next 25 residues after the first ~50 amino acids are found within a high-order region in both DDX39A and DDX39B. This result does not provide evidence for the different responses of DDX39A and DDX39B to heat shock. Descriptions in lines 111-114 do not support descriptions in lines 114 – 118.

We thank the Reviewer for their comment, as it has allowed us to clarify the text in this section. To address the Reviewer’s first two comments, we have made substantial revisions to the text describing the role of amino acids 50-75 in DDX39 aggregation. We hope that the text specifically pertaining to this point has addresses the Reviewer’s concerns:

Line 147: *“A more probable explanation for the effects of amino acids 50-74 on A-body stress-specificity, is that this region is important for the proper folding and tertiary structure of this protein. IUPred3 analysis and AlphaFold predictions (AF-Q13838-F1) support this notion, as the first ~50 residues of DDX39B encode an intrinsically disordered domain, while the next 25 amino acids are part of the N-terminal ordered structure (Figure 2D). Thus, we predict that protein structure could be a critical determinant in heat shock-specific A-body recruitment.”*

3) For the molecular dynamic simulation in Fig.6C, besides the difference observed around I153, there is also a big difference for residue 60~70. Does it mean some long-range associations between residues around I153 with residues 60~70? The manuscript does not describe the striking difference for MD simulation for residues 60~70, which seems important to prevent the aggregation of DDX39B.

At the request of another Reviewer, we replaced the previous molecular dynamics analyses (Reviewer 3, comment 13).

The original data used large temperature differences (27°C and 52°C) to amplify structural signals and reduce the ambiguity in dynamic trends typically seen in proteins when looking at small temperature differences. Despite the potential issues of using a narrow temperature range, our new analyses at physiological temperatures (37°C and 43°C) clearly demonstrate differences in the thermo-stability of DDX39A and DDX39B (**new Figure 6**). We have also re-focused the data to highlight changes in dynamics (RMSF) between DDX39A and DDX39B at 43°C (**new Figure 6C**), instead of the previous data that looked at changes in dynamics of DDX39A or DDX39B when temperatures increased from 27°C and 52°C

(old Figure 6C). We believe this is a more relevant analysis, as the crux of our work is a comparison between how these proteins act under heat shock conditions.

Additionally, the data in **new Figure 6C** shows a more modest difference at amino acids 60-70, but a major increase in dynamics around the F184 hydrophobic pocket. This aligns well with our proposed model.

4) The authors suggest that thermal stability determines the heat shock-specific sequestration. Can the authors characterize the thermal stability of DDX39A and DDX39B (or relative mutant/chimera)? MD simulation of DDX39A is based on a predicted structure, while DDX39B's modeling is based on a crystal structure. It would be more convincing if the authors could verify the changes in protein thermal stability using purified protein in experiments.

We previously spent almost 18 months trying to purify soluble DDX39A and DDX39B from bacterial and mammalian sources, with no success. In fact, this was the reason we turned to molecular dynamics to provide additional validation to our cellular observations. Part of the struggles purifying proteins are highlighted in the **new Figure 1C**, as DDX39A and DDX39B form prominent inclusion bodies upon expression in bacteria (this is a model of amyloid aggregation).

However, we have recently developed an assay to obtain more direct biophysical evidence on the thermo-sensitivity of the proteins using soluble supernatants from lysed, untreated mammalian cells. In these experiments, we first pre-cleared the lysate of aggregates and cellular debris by centrifugation (**new Figures 1G and 7C**). We proceeded to heat lysates to 43°C (1hr), then run the samples on non-denaturing- and denaturing-PAGE. Despite the uniform presence of the proteins in the lysates (**Figure 1G/7C, SDS-PAGE: Total**), monomeric DDX39B/hnRNPA1 could be seen at 43°C, while DDX39A/hnRNPA0 monomers disappeared at the elevated temperature (**Figure 1G/7C, Native-PAGE**). Taking aliquots of these incubated lysates, we centrifuged down any temperature-induced aggregates, ran the pellets on an SDS-PAGE, and found direct exposure to 43°C pushed DDX39A/hnRNPA0 into an aggregated state (**Figure 1G7C, SDS-PAGE: Pellet**).

We took this approach one step further and tried the experiment using the central domain (aa 100-250) swaps, which change the A-body targeting phenotypes of DDX39A and DDX39B. This result clearly showed that thermo-sensitive aggregation was imparted or abrogated by the presence of the DDX39A or DDX39B 100-250 regions, respectively (**Figure 1G**). We hope the Reviewer will appreciate the difficulty in working with these extremely temperature-sensitive and aggregation prone proteins and finds this additional data as valuable as we do.

5) Line 199, what is the residue numbering for the N-term lobe?

We thank the Reviewer for the comment, and we have revised the text to ensure the Reader is clear on the regions of DDX39 being analyzed in each simulation:

Line 232: “During the simulation the average root mean square deviation (RMSD) of the ordered regions of the N-terminal lobes of DDX39A and DDX39B (amino acids 46-250) were similar at 37°C (Figure 6A, 6B-top panel).”

6) Amyloid fibril is structurally dominated by β -sheet structure. What proteins construct the β -sheet amyloid fibril in the amyloid-like body? For example, do amyloidogenic sequences predicted by AmylPred2 in DDX39 form amyloid fibril *in vitro*?

This is a very interesting question. We acknowledge that immobilization and insolubilization are hallmarks of proteins adopting an amyloid-like conformation as they enter the A-bodies, but they are not definitive proof. Current technology makes it impossible to provide direct *in cellulo* evidence that a particular protein is in an amyloid state. However, using tools we developed in our previous work (Audas, *et al.*, 2016, *Developmental Cell*) we can increase our confidence that DDX39A and DDX39B are capable of adopting an amyloid-like conformation.

Bacterial inclusion bodies are a model of amyloid aggregation, as proteins within these structures have been shown to form amyloid fibrils (de Groot, *et al.*, 2009, *TIBS* and Wang, *et al.*, 2008, *PLoS Biology*). We expressed DDX39A and DDX39B in a bacterial setting and found that both proteins could generate inclusion bodies that had an affinity for the amyloidophilic dye Congo red, similar to the quintessential pathological amyloid aggregate β -amyloid (1-42). As expected, GFP alone does not aggregate (new Figure 1C).

We also took the Reviewer's advice (synthesizing peptides that were predicted to fibrillate *in vitro*) and ran a Thioflavin T fibrillation assay (new Figure 2C and S3D). In our hands, the DDX39A and DDX39B peptides that were predicted by

AmylPred2 to fibrillate could form amyloid aggregates *in vitro*. Together, these data suggest that DDX39A and DDX39B have the capacity to adopt an amyloid-like state, and increases our confidence that they do so within the A-bodies. We also included text discussing this in the manuscript:

Line 83: “For these RNA helicases, the putative adoption of an amyloid-like conformation within A-bodies was demonstrated by the hallmark shift towards the insoluble (Figure S1B) and immobile (Figure S1C-D) biophysical state and the inherent capacity of these proteins to generate fibrils, when expressed in a bacterial inclusion body assay (Figure 1C: a model setting of amyloid aggregation^{21,39,40}).”

Line 132: “As AmylPred2⁴² predicts that each of these regions contain aggregation prone clusters (Figure S3B-C – Bottom Panel), we synthesized DDX39 peptides and ran a Thioflavin T (ThT) fibrillation assay to determine whether DDX39 fragments could adopt an amyloid conformation. In this assay, peptides from the central region of DDX39A (160-199) and DDX39B (161-200) generated ThT-positive amyloid fibrils at a significantly higher rate than AmylPred2-negative DDX39A (1-39) and DDX39B (1-40) regions (Figure 2C, S3D).”

Reviewer #3:

Amyloid bodies (A-bodies), an RNA-dependent subnuclear condensate, are an example of functional amyloid which forms due to the cellular stress response. However, little is known regarding the mechanism that underlies the production of physiological amyloid assemblies. In this manuscript, the authors have proposed a model where the tertiary structure of the proteins, along with their thermal stability, collectively dictates their selective aggregation within amyloid bodies (A-bodies). Here, the authors have made a library of mutations in A-body-related proteins (DDX39 and hnRNPA) to probe the mechanism that regulates heat shock-mediated protein recruitment within the A-bodies and their further aggregation. Although the set of related proteins has enormous similarities in their predicted structures, sequences, and propensity to form aggregates, however, they show differential recruitment inside A-bodies when exposed to increased temperature. Based on their rigorous mutation and A-bodies targeting study, the authors have determined a crucial structural element that regulates heat-shock-specific amyloid aggregation, which can either be induced or restricted by manipulating their structural pockets. The study also concludes that the intrinsically disordered regions did not contribute to the A-bodies targeting property, which is a fascinating outcome of the manuscript.

The experiments were well thought out and executed elaborately with adequate discussion. The manuscript mainly focuses on preparing a mutation library and A-bodies targeting study, which is considerable work. However, the paper lacks a discussion on the in-vitro characterization of A-bodies formed in different conditions. Moreover, the microscopy images throughout the manuscript are not up to the journal's standard. I have the following questions.

We thank the Reviewer for their comprehensive assessment of our manuscript and echo their opinion about the fascinating nature of the mechanism we have uncovered. We have invested considerable effort in improving the quality of the microscopy data throughout the manuscript and hope the Reviewer will now agree that our work is up to the journal's high standards. Overall, we thank the Reviewer for their feedback, as we believe it has significantly enhanced the quality of our work.

1. The introduction does not explain the role of two sets of model proteins used in this manuscript, DDX39 and hnRNPA. The authors should briefly describe these two proteins highlighting their relevance and functionality for A-body formation.

We have added the following text to the introduction to highlight the basic function and relevance of the proteins within this cellular stress response pathway:

Line 59: “DDX39A and DDX39B are members of the DEAD box family of RNA helicases, which have been implicated in genomic integrity³⁴, RNA splicing³⁵, and mRNA export³⁶, while the heterogeneous nuclear ribonucleoprotein A0 (hnRNPA0) and A1 (hnRNPA1) have been linked to multiple aspects of RNA biogenesis³⁷. Thus, stress-specific recruitment of these proteins could significantly alter genome stability and RNA metabolism, providing a site of cellular regulation that tailors this stress-response pathway to different environmental perturbations.”

2. In the introduction section, at the end, the author should briefly discuss their main finding and conclusion. The introduction section looks abruptly ends without any conclusion.

We have adjusted the introduction to include a brief description of the conclusions:

Line 64: “Our work demonstrates that individual proteins can possess thermo-sensitive structures that act as direct temperature sensors to mediate A-body recruitment and aggregation. Interestingly, the intrinsically disordered domains of these proteins did not contribute to this stress-specific aggregation mechanism, as the highly ordered structures possessed the thermo-sensing functionality. Collectively, these data highlight a new post-translational regulatory mechanism, where physiological amyloid aggregation can be rapidly and specifically controlled by the thermo-sensitivity of the tertiary structure of individual cellular proteins.”

3. The author has mentioned in line 65 (also illustrated in Figure 1A) that DDX39A and DDX39B show 90% amino acid sequence identity. The authors should indicate the sequence homology with sequence alignment study and incorporate that in the supplementary information.

We had originally included a DDX39A-DDX39B sequence alignment (amino acids 100-250) in Figure 3A, and a schematic of the locations where amino acids diverge in Figure 1C. However, we agree with the Reviewer that an alignment of the full-length proteins would help Readers understand the conservation of these proteins, as well as clarify the confusing residue numbering that the Reviewer highlights in comment 10.

A new Figure S1A has been included in the manuscript.

4. In Figure 1B, MCF-7 cells have been transfected with DDX39A-GFP, DDX39B-GFP, and the amyloid body marker protein CDC73-mCherry. Further, cells were left untreated or exposed to heat shock or extracellular acidosis. Though the cells treated with heat shock were stained with Thioflavin S (ThS) dye, the cells treated with extracellular acidosis were not stained with ThS dye. The authors should also show the ThS result for the acidosis study. Moreover, this figure does not clearly show the co-localization between the A-bodies marker and DDX39 proteins. Authors should provide better images for co-localization and can also calculate the percentage of co-localization based on their intensity profile. Another question that strikes in this figure is as follows. The cells were exposed to heat shock for 4 hrs at 43 °C. Is there any reason or reference for that specific temperature and time duration utilized for heat shock? If yes, the authors should clarify that. Otherwise, the author should show and incorporate the optimization study in the supplementary section.

We appreciate the Reviewer's feedback on these points.

- (1) In a revised version of Figure 1B, we have added Thioflavin S staining to the acidotic samples to remind Readers that both stressors generate A-bodies with amyloid-like properties (Audas, *et al.*, 2016, *Developmental Cell*).
- (2) To highlight the co-localization of DDX39A and the absence of DDX39B (within the heat shock-induced A-bodies), we have taken high-resolution images of cells and calculated the signal intensity of these proteins and the A-body marker molecule CDC73. As can be seen in the new Figure S1E, DDX39A efficiently co-localizes with CDC73 in the A-bodies, while DDX39B does not.
- (3) In our previous work (Lacroix, *et al.*, 2021, *Journal of Cell Science*), we measured the temperature and pH at which A-bodies form in a cross-section of eukaryotic species. This work highlights the need for severe (but survivable) stress exposure to induce A-body formation. Overall, the temperatures, pH, and time of the A-body inducing stimuli are in line with previous work (Audas, *et al.*, 2012, *Molecular Cell*, Audas, *et al.*, 2016, *Developmental Cell* and Wang, *et al.*, 2019, *Cell Reports*).

5. Figure 1C shows Western blots for the range of substitution constructs of DDX39 expressed in MCF-7 cells in untreated or heat-shock-treated cells. Here, the blots lack loading control for both the cell lysate and insoluble fraction. The author should show the gel image with loading controls GAPDH and Histone H3 for soluble and insoluble fractions in the supplementary section for a fair comparison.

GAPDH and Histone H3 were not used as loading controls for these experiments, as they were intended to demonstrate that the insoluble fraction (Histone H3) had been extracted from the cell lysate (GAPDH a soluble protein).

We cannot use these samples as a loading control chiefly due to the variability in the transfection efficiency of each construct. For the calculations presented in this figure (moved to **Figure S2F**), the loading control is the chimeric DDX39 protein in the heat shock cell lysate fraction (i.e., the sample that the insoluble fraction was extracted from). To demonstrate that insoluble fractions were generated in the heat shock samples, we have included a **new Figure S2G**, which shows the presence/absence of DDX39A and DDX39B constructs in the GAPDH and Histone H3 verified fractions.

F

G

6. Figures 1E, 7H, S2 B, S3 A and S3 C, S5 C Quantification of FRAP Data have been illustrated without normalization. The FRAP data can better be represented as normalized intensity (with positive and negative background correction) and data fitting for better comparison.

We thank the Reviewer for their helpful advice. We have normalized the data in all the FRAP figures (**Figure 1F, 7I, S1C, S1D, S4A, S6C, and S7D**), and a representative quantification is displayed to the right (**Figure 1F**, which replaced Figure 1E).

7. In Figure 2A, A-body targeting efficiency has been calculated as the average A-body pixel intensity relative to the average nuclear background pixel intensity. Since a significant amount of work is based on the quantitative microscopy approach, the authors should elaborate on the calculation from the microscopy for at least one representative data set in the supplementary section for a better understanding of the calculation.

While this methodology has been used in several papers, we agree with the Reviewer that Readers would be aided by an example of this quantitative approach.

We have added a panel (new Figure S2E) to describe the calculations and demonstrate how the data is collected.

E

Relative Signal Intensity =
$$\frac{\left[\frac{(\text{A-body} - \text{Back})}{1} + \frac{(\text{A-body} - \text{Back})}{2} \right]}{(\text{Nucleus} - \text{Back})} \div 2$$

8. In line 98, the authors have demonstrated via a quantitative microscopy approach that GFP is sequestered by the amino acids 75-250 proteins when exposed to heat shock conditions. In this case, the author should confirm the statement by utilizing an A-body marker and co-localization experiment to confirm whether the GFP-tagged proteins co-localize with the A-body marker.

We have included this data as the new Figure S3A, where we show targeting and co-localization of DDX39B (75-250) within the CDC73-positive A-bodies.

9. Figure 2C: IUPred3 prediction maps of disordered protein regions for DDX39A and DDX39B proteins have been shown. The authors should also mention the amino acid domains and the IUPred prediction graph, which will provide domain-specific structural information.

We are unclear about what the Reviewer is requesting here, but we have added established RNA helicase domains for DDX39 to the top of the IUPred3 prediction in Figure 2D.

This domain map corresponds to the sequence information in the new Figure S1A (comment 3). We hope this addresses the comment.

10. Figure 3B and 3D quantify the A-bodies targeting efficiency of point mutations that replace each of the unique residues in DDX39A with their DDX39B equivalents and vice versa. However, the amino acid positions in Figure 3B and the corresponding reciprocal mutations in Figure 3D are different by one unit throughout all the variants (for example, F184L in Figure 3B and L185F in Figure 3D). Am I missing something in this figure? The author should recheck the designated position of point mutations and clarify the concern.

Unfortunately, there is one extra amino acid in the N-terminus of DDX39B. This means that all of the substitutions appear to be off by one residue. In response to the Reviewer's third comment, we added a sequence alignment for the DDX39A-DDX39B sequences (**new Figure S1A**). Hopefully, by highlighting the extra residue in red, we will help future Readers note the differences in the DDX39 numbering and alleviate this confusion. We have also included text explicitly describing this numbering issue in the results section:

Line 167: *“As predicted, the single reciprocal mutations L185F and V224C (DDX39B contains one extra residue at amino acid 18 [Figure S1A], hence the different residue position) ...”*

11. In Figure 3B and 3D, the authors have mentioned that “the substitution of F184L or C223V reduced A-body localization and increased mobility of the mutant DDX39A proteins to levels similar to that of the DDX39B” thus exerting a “masking effect” on DDX39A. If these two single-point mutations pose the masking effect on the protein, then it is expected that during the reciprocal substitution where amino acids L from DDX39B is substituted to amino acid F from DDX39A would release the masking effect with an enhancement in the A-bodies targeting efficiency. But the representative data in Figure 3D shows the opposite of the expected data. The author should clarify the apparent discrepancy in the representative data compared to that of the expected data.

Here we were attempting to highlight the observation that both the F184L and C223V mutations independently exert an inhibitory effect, suggesting that the presence of either is sufficient for an A-body negative phenotype (i.e., a dominant-inhibitory effect on aggregation). This led to our prediction that introducing only one reciprocal mutation into DDX39B would be inadequate to induce its A-body targeting, as it still has a "dominant-inhibitory" mutation at the other position.

We thank the Reviewer for pointing out the confusing language used, and we changed the phrasing in the text to convey our thinking more clearly to the Reader.

Line 155: *“Of the 17 sites that encoded different amino acids within the 100-250 region, multiple substitutions significantly repressed A-body recruitment (Figure 3B). The F184L and C223V substitution were the most prominent, as they could independently reduce A-body localization (Figure 3B-C) and increase DDX39A mobility (Figure S4A) to DDX39B levels. Under acidotic conditions neither F184L nor C223V had any effect on full-length DDX39A sequestration (Figure 3C and Figure S4B), and their incorporation into the minimal DDX39A (99-249) constructs also failed to repress A-body recruitment under both heat shock and acidotic conditions (Figure S4C). This suggests that DDX39B amino acid substitutions are not disrupting the generic acidosis/heat shock A-body targeting motif(s). Next, we sought to perform the reciprocal experiments and impart the heat shock A-body targeting properties of DDX39A on DDX39B. We noted that both F184L and C223V substitutions independently inhibit DDX39A recruitment (Figure 3B-C), suggesting that these residues exert a dominant-negative effect on this form of heat-induced protein aggregation. Therefore, we expected the presence of either residue in DDX39B would prevent its aggregation. As predicted, the single reciprocal mutations L185F and V224C (DDX39B contains one extra residue at amino acid 18 [Figure S1A], hence the different residue position) failed to induce A-body targeting in DDX39B, as each construct possessed an inhibitory residue at the other established site (Figure 3D).”*

12. In Figure 4D, the Author should show the individual effects of the three mutations (I106L+N188H+V167I) other than the T114S mutation as well to precisely depict the effect of individual mutation on the relative signal intensity from A-body.

We are not entirely sure we have correctly interpreted the Reviewers question, but we think they are asking for the effects of the I106L, V167I, and N188H substitutions alone (i.e., not in combination with the T114S substitution).

If that is the case, we'd like to direct the Reviewer to Figure 3B, where these mutations are presented (green highlighting is not included in the final manuscript). Each of these constructs reduced the relative signal intensity from the A-bodies.

We have also amended the text to make it easier for the Reader to find this data:

Line 194: “*Individually, each of these residues appear to impair DDX39A sequestration (Figure 3B), however, the DDX39A (quadruple; T114S+I106L+N188H+V167I) construct repressed heat shock-induced A-body targeting to wild-type DDX39B levels (Figure 4D) ...*”

13. In Line 195, the authors have investigated the root mean square deviation (RMSD) calculation of DDX39A by using 52 °C as high temperature and 27 °C as low temperature. The authors should clarify the basis for choosing these specific temperatures.

The original data used large temperature differences (27°C and 52°C) to amplify structural signals and reduce the ambiguity in dynamic trends typically seen in proteins when looking at small temperature differences. This is a typical strategy of many molecular dynamics simulations.

However, we sincerely thank the Reviewer for pushing us on this point! Despite our initial concerns, our new analyses at physiological temperatures (37°C and 43°C) clearly demonstrate robust and marked differences in the thermostability of DDX39A and DDX39B (**new Figure 6**).

Re-visiting this work has also allowed us to re-focus the data, highlighting changes in dynamics (RMSF) between DDX39A and DDX39B at 43°C (**new Figure 6C**), instead of the previous data that looked at changes in dynamics of DDX39A or DDX39B when temperatures increased from 27°C and 52°C (old Figure 6C). We believe this is a more relevant analysis, as the crux of our work is a comparison between how these proteins act during high temperature conditions.

14. Figure 7F shows heat shock-treated MCF-7 cells expressing the A-bodies marker CDC73-mCherry and hnRNPA1 (F34A)-GFP. Even though the insets display the CDC73-mCherry signal, the authors should provide a merged image for a better demonstration of the co-localization.

We have re-made the image in Figure 7F (right) to better highlight the co-localization of CDC73-mCherry and hnRNPA1 (F34A)-GFP within the A-bodies.

We have also done this for all other A-body containing images throughout the manuscript.

15. In Figures 7G, S1 A, and E, in both the Western blots, it is quite evident from the blot that the GAPDH expression is inconsistent throughout the lane (more inconsistent in S1 E). In this case, the band's intensity should be calculated, analyzed, and compared.

We repeated the blot in Figure S1E (now **Figure S2D**) and frequently observe more GAPDH in the heat shock treated samples (relative to no treatment or acidosis). As discussed above (comment 5), GAPDH and Histone H3 are not loading controls for this experiment, they are soluble/insoluble controls to demonstrate that the fractionation worked.

The loading control for this experiment would be the intensity of the DDX39 protein in the corresponding cell lysate, and using this we can see that there is a clear difference in the abundance of insoluble DDX39A and DDX39B under heat shock conditions.

16. Figure S2 C shows heat-shock-treated MCF-7, A549, and HEK293 cells expressing DDX39A-GFP and DDX39B-mCherry. This experiment illustrated that the cell line is not responsible for mediating the divergence in stress-specific amyloid body targeting. However, a question that arises here is why the cells were only heat-shock treated and not exposed to acidosis. The author should clarify the point.

For the sake of space, we had only included the heat shock treated samples, since this condition was the only one where there was a divergence in the A-body localization for the DDX39A and DDX39B proteins. At the Reviewer's request we included data for all the treatment conditions in A549 and HEK293 (**new Figure S1E**). We have also co-expressed the marker molecule CDC73 to confirm the translocation of these proteins to A-bodies. We agree with the Reviewer that this data gives the Reader a more complete understanding of the targeting profiles of these proteins in multiple cell lines.

17. Figure S6: Please review the figure, as the figure legends do not match the illustrated data.

We thank the Reviewer for pointing this out. The Figure Legend has been corrected.

18. The authors should perform immunostaining with amyloid-specific OC antibody to confirm the amyloid state of the A-bodies formed in cells when exposed to heat shock and acidosis. Moreover, the authors should also perform electron microscopy of the A-bodies isolated from cells to visualize their morphology. Apart from Thio-S staining, I don't see any effort from the author to characterize the A-body for the amyloid formation.

Unfortunately, due to space limitations we can't repeat our previous experiments demonstrating the amyloid-like nature of the A-bodies within this manuscript (Audas, *et al. Developmental Cell*, 2016). We have selected several of the most efficient hallmarks (insolubilization, immobilization, and affinity for amyloid dyes), but we hope that the Reviewer and future Readers will be satisfied by examining our previous publication.

In Audas, *et al. Developmental Cell*, 2016, each of the requested experiments were performed on cells exposed to the same conditions used in this paper. There, we stained A-bodies with the OC antibody (Figure 1G and S1H) and performed electron microscopy (Figure 1F, 1G, and S1G). We also used electron microscopy on fragments of an A-body constituent VHL (Figure 4I) to demonstrate that it could adopt an amyloid conformation *in vitro*. If the Reviewer is interested, there are also other assays in this publication that demonstrate the amyloid characteristics of these foci and their constituents, including insolubilization/immobilization assays, proteinase K-resistance, X-ray diffraction, inclusion body formation, and staining with 6 distinct amyloidophilic dyes. We hope the Reviewer will agree that this exhaustive characterization does not need to be repeated in this manuscript.

19. Apart from the in-cell studies performed in the manuscript, which are indeed rigorously done, the authors should also perform in-vitro studies to characterize the A-body aggregates through biophysical experimentation (e.g., CD, Thioflavin T assay, etc.). I would encourage the author to do in vitro experiments with DDX39A and B and a few mutants for their thermal response to the structural changes (secondary or tertiary structure) and aggregation propensity (using ThT fluorescence, CD, and electron microscope). That might correlate with the cellular data, establishing the structure-function (A-body formation here).

We previously spent almost 18 months trying to purify soluble DDX39A and DDX39B from bacterial and mammalian sources, with no success. In fact, this was the reason we turned to molecular dynamics to provide additional validation to our cellular observations. Part of the struggles purifying these proteins are highlighted in the **new Figure 1C**, as DDX39A and DDX39B form prominent inclusion bodies upon expression in bacteria (this is a model of amyloid aggregation). We were also able to synthesize peptides for regions of DDX39A/B and perform Thioflavin T fibrillation assays (**new Figure 2C, S3D**), to further demonstrate the amyloidogenic capacity of these RNA helicases.

Recently, we developed an assay to obtain more direct biophysical evidence on the thermo-sensitivity of the proteins using soluble supernatants from lysed, untreated mammalian cells. In these experiments, we first pre-cleared the lysate of aggregates and cellular debris by centrifugation (**new Figures 1G and 7C**). Next, we heated the lysates to 43°C (1hr), and ran the samples on non-denaturing- and denaturing-PAGE. Despite the uniform presence of the proteins in the lysates (**Figure**

1G/7C, SDS-PAGE: Total), monomeric DDX39B/hnRNPA1 could be seen at 43°C, while the DDX39A/hnRNPA0 monomers disappeared at the elevated temperature (**Figure 1G/7C, Native-PAGE**). Taking aliquots of these incubated lysates, we centrifuged down any *in vitro* temperature-induced aggregates, ran the pellets on an SDS-PAGE, and found direct exposure to 43°C pushed DDX39A/hnRNPA0 into an aggregated state (**Figure 1G7C, SDS-PAGE: Pellet**).

As the Reviewer requested, we tried the experiment using the central domain (aa 100-250) swaps, which change the A-body targeting phenotypes of DDX39A and DDX39B. This result clearly showed that thermo-sensitive aggregation was imparted or abrogated by the presence of the DDX39A or DDX39B 100-250 regions, respectively (**Figure 1G**). We hope the Reviewer will appreciate the difficulty in working with these extremely temperature-sensitive and aggregation prone proteins and finds this additional data as valuable as we do.

REVIEWERS' COMMENTS

Reviewer #1 (Remarks to the Author):

The authors did a nice job of addressing my concerns with newly added data and improved data analysis. I have no further concerns.

Reviewer #2 (Remarks to the Author):

The revised manuscript is significantly improved. Most of my comments have been very well addressed.

There are a few additional concerns:

1) Line 151: 'Thus, we predict that protein structure could be a critical determinant in heat shock-specific A-body recruitment.' Here, 'protein structure' is very vague. 'thermal stability of the tertiary structure' seems better and more precise.

2) Line 355-356: In the last sentence, '...for the quick and reversible inactivate cellular pathways in an environmental context dependent manner'. Is the A-body formation 'reversible' when the heat-shock condition is removed? Amyloid is usually very stable, and it is usually very difficult to reverse the process of Amyloid formation because of the high energy barrier. I didn't see the evidence in the manuscript showing that the A-body formation is reversible. Please comment on this point in the discussion. And if the A-body is not reversible, then, how can we conclude that this is a physiological/functional amyloid aggregation and not just a one-way deadly end aggregation process?

Reviewer #3 (Remarks to the Author):

The manuscript entitled "Protein Thermal Sensing Regulates Physiological Amyloid Aggregation" has been significantly improved after the revision. The authors have addressed all of our concerns

satisfactorily. Moreover, they have also performed several additional experiments to support their justification, which is quite impressive. I think the manuscript is now suitable for publication.

Reviewer #1:

The authors did a nice job of addressing my concerns with newly added data and improved data analysis. I have no further concerns.

We thank the Reviewer for their time and constructive feedback. The changes that they have recommended have substantially improved the quality of our work.

Reviewer #2:

The revised manuscript is significantly improved. Most of my comments have been very well addressed.

We appreciate all the Reviewer's feedback and would like to thank them for helping us improve our research and strengthen our findings.

There are a few additional concerns:

1) Line 151: 'Thus, we predict that protein structure could be a critical determinant in heat shock-specific A-body recruitment.' Here, 'protein structure' is very vague. 'thermal stability of the tertiary structure' seems better and more precise.

We have revised the text as the Reviewer recommended. It now reads:

"Thus, we predict that thermal stability of the tertiary structure could be a critical determinant in heat shock-specific A-body recruitment."

2) Line 355-356: In the last sentence, '...for the quick and reversible inactivate cellular pathways in an environmental context dependent manner'. Is the A-body formation 'reversible' when the heat-shock condition is removed? Amyloid is usually very stable, and it is usually very difficult to reverse the process of Amyloid formation because of the high energy barrier. I didn't see the evidence in the manuscript showing that the A-body formation is reversible. Please comment on this point in the discussion. And if the A-body is not reversible, then, how can we conclude that this is a physiological/functional amyloid aggregation and not just a one-way deadly end aggregation process?

We are a little surprised that we forgot to highlight the fascinating fact that this form of amyloid aggregation is rapid and reversible in our manuscript. Data demonstrating this can be found in our previous papers (Audas *et al.* 2016. *Developmental Cell* and Lacroix *et al.* 2021. *Journal of Cell Science*). To clarify this point for the readers, we have amended two sentences in the Introduction to highlight these previous findings:

*"Biogenesis of A-bodies occurs in response to elevated temperature or low environmental pH, **while a subsequent return to normal growth conditions results in their disassembly.**"*

*"As these **reversible** amyloids have been found in species from across the eukaryotic domain,..."*

Reviewer #3:

The manuscript entitled “Protein Thermal Sensing Regulates Physiological Amyloid Aggregation” has been significantly improved after the revision. The authors have addressed all of our concerns satisfactorily. Moreover, they have also performed several additional experiments to support their justification, which is quite impressive. I think the manuscript is now suitable for publication.

We thank the Reviewer for their comments throughout the review process. Their insight and suggestions have allowed us to generate a more solid and impactful manuscript.